# The Multiple Aperture SAR Interferometry (MAI) Technique for the Detection of Large Ground Displacement Dynamics: An Overview

**Pietro Mastro** [1] , **Carmine Serio** [1] , **Guido Masiello** [1] and **Antonio Pepe** [2,*]

[1]  The University of Basilicata, School of Engineering, 85100 Potenza, Italy; pietro.mastro@unibas.it (P.M.); carmine.serio@unibas.it (C.S.); guido.masiello@unibas.it (G.M.)

[2]  National Research Council of Italy, Institute for the Electromagnetic Sensing of the Environment (CNR-IREA), 80124 Napoli, Italy

*  Correspondence: pepe.a@irea.cnr.it

**Abstract:** This work presents an overview of the multiple aperture synthetic aperture radar interferometric (MAI) technique, which is primarily used to measure the along-track components of the Earth's surface deformation, by investigating its capabilities and potential applications. Such a method is widely used to monitor the time evolution of ground surface changes in areas with large deformations (e.g., due to glaciers movements or seismic episodes), permitting one to discriminate the three-dimensional (up–down, east–west, north–south) components of the Earth's surface displacements. The MAI technique relies on the spectral diversity (SD) method, which consists of splitting the azimuth (range) Synthetic Aperture RADAR (SAR) signal spectrum into separate sub-bands to get an estimate of the surface displacement along the azimuth (sensor line-of-sight (LOS)) direction. Moreover, the SD techniques are also used to correct the atmospheric phase screen (APS) artefacts (e.g., the ionospheric and water vapor phase distortion effects) that corrupt surface displacement time-series obtained by currently available multi-temporal InSAR (MT-InSAR) tools. More recently, the SD methods have also been exploited for the fine co-registration of SAR data acquired with the Terrain Observation with Progressive Scans (TOPS) mode. This work is primarily devoted to illustrating the underlying rationale and effectiveness of the MAI and SD techniques as well as their applications. In addition, we present an innovative method to combine complementary information of the ground deformation collected from multi-orbit/multi-track satellite observations. In particular, the presented technique complements the recently developed Minimum Acceleration combination (MinA) method with MAI-driven azimuthal ground deformation measurements to obtain the time-series of the 3-D components of the deformation in areas affected by large deformation episodes. Experimental results encompass several case studies. The validity and relevance of the presented approaches are clearly demonstrated in the context of geospatial analyses.

**Keywords:** SAR interferometry (InSAR); multiple-aperture SAR interferometry (MAI); along-track/across-track displacement; spectral diversity (SD); atmospheric path delay; ionosphere; water vapour; TOPS

---

## 1. Introduction

The exploitation of Earth observation (EO) methods, based on the use of instruments operating in the microwave region of the electromagnetic spectrum, represents a common practice nowadays in the scientific community [1–6]. In this context, the development of new remote sensing techniques and the consolidation of well-established ones is fostered by increasing amounts of EO data collected by

several sensors (on-board space or aerial vectors) that have emerged in recent years. A significant role is performed by the technologies based on their use of synthetic aperture radar (SAR) data that allow long-lasting and extensive monitoring of the Earth's surface. SAR sensors are active instruments [7] that have been extensively employed for studying and monitoring the Earth's surface [8–10], as well as the surfaces of other celestial bodies such as the solar system planets [11–13]. SAR instruments exploit the interaction of packets of electromagnetic waves emitted by RADAR sensors that are backscattered from objects within an observed area. The signals used by radar sensors usually lie in the microwave band (i.e., in the range of wavelengths between 1 cm and 1 m). This means that radar instruments can be used efficiently under any meteorological condition (i.e., in the presence of a significant cloud cover) with a full day-and-night operational capability. In this framework, several techniques have been developed to process and analyze SAR data. Among these, in recent decades, SAR interferometry techniques (InSAR) [14–16] have gained a role of particular relevance in the study of the Earth's surface displacement phenomena. Ground deformations can be induced by numerous natural phenomena such as broad fractures due to big earthquakes [17–22], slow-moving landslides [23–25], eruptions of active volcanoes [26–30] and glacier movements [31,32]. Human beings also severely contribute to changes in the environment. In this framework, InSAR can monitor changes in highly urbanized zones due (for instance) to ground-water extraction as well as the deterioration of buildings and private and public facilities [33–39]. InSAR can monitor these modifications over time with high accuracy, allowing the retrieval of useful information for world governments to mitigate the risks for people and the environment. However, one of the main limitations of the InSAR methodology is that only sensor-to-target line-of-sight (LOS) projections of ground deformation can be detected and analyzed. Nevertheless, the combination of complementary information obtained by multi-satellite/multi-track InSAR observations can allow the discrimination of overall ground deformations in 3-D profiles (up–down, east–west, north–south). However, almost all modern spaceborne satellites fly along near-polar orbits (i.e., parallel to the north–south direction). This condition poses a problem in discriminating the north–south components of ground deformations measured from LOS-projected deformation signals, which have a low sensitivity to north–south displacements. Such a problem has been overcome in recent years through the application of pixel-offset (PO) tracking [40] and multiple aperture SAR interferometry (MAI) [41] approaches, which both permit computing the dynamics of large along-track ground surface changes. PO techniques rely on the measurement of local sub-pixel offsets between a couple of amplitude SAR images. As a consequence, the accuracy of estimating the azimuth/range offsets with amplitude matching can only reach a fraction (i.e., 1/30th) of the pixel spacing, which means the smaller the pixel size, the higher the accuracy. In a practical sense, considering the first-generation SAR satellites (e.g., the Advanced Synthetic Aperture Radar (ASAR), mounted onboard the Envisat platform /ASAR), PO can reach accuracy values that are at best greater than 10 cm. PO methods initially were mostly applied to studying single deformation events [32,40,42–46], but recently some efforts have also been made to extend these techniques to investigating SAR Doppler anomalies in focused SAR data [47] and for the generation of 3-D ground displacement time-series [48]. The latter is obtained by combining the information derived from both sequences of multi-temporal DInSAR interferograms and amplitude maps (e.g., [48]). A substantial improvement in measuring along-track displacement has been made with the MAI technique [41], which exploits spectral diversity (SD) processing [49], consisting of splitting the azimuth (range) SAR signal spectrum into separate sub-bands, to get an estimate of the surface displacement along the azimuth (sensor line-of-sight) direction. In particular, the larger the spectral separation between sub-bands, the higher the accuracy of the attainable measurements. Splitting band approaches were initially exploited by the oceanography community to infer direct measurements of ocean surface velocity, analyses and interpreting the Doppler shift between two RADAR SAR images acquired over the ocean [50,51]. The accuracy of the PO and MAI methods has been assessed and cross-examined in several studies [41,52,53]. The accuracy of MAI measurements has had a range between 3 and 7 cm, depending on the SAR sensor used, processing parameters and coherence of the MAI. Considering

the parameters of an L-band system (e.g., the Phased Array type L-band Synthetic Aperture Radar (PALSAR 1-2), mounted onboard the Advanced Land Observing Satellite (ALOS) 1-2 platforms ALOS-PALSAR), the MAI technique can reach an accuracy of about 3–4 cm. MAI/PO techniques can complement results obtained using advanced multi-temporal InSAR approaches for the generation of ground deformation time-series [54–64]. A few solutions for acquired 3-D deformation time-series by combining multi-satellite/multi-observation (ascending/descending passes) data tracks have been proposed in the literature [65–75]. Among these, one method called Minimum Acceleration (MinA) combination [75,76] allows the generation of 2-D (3-D) displacement time-series of the Earth's surface. This is achieved by combining LOS-projected deformation time-series obtained from different SAR platforms by imposing that the reconstructed 2-D (3-D) components of the ground deformation have a minimum acceleration. Noteworthy, the SD approaches are also currently used extensively to improve the performance of co-registration of the SAR images acquired with the novel Terrain Observation with Progressive Scans (TOPS) mode [77–81], which is the acquisition mode of the interferometric wide (IW) Sentinel-1 data [82]. Moreover, SD methods are used to estimate and correct atmospheric phase artefacts [83–87] that corrupt InSAR-driven ground deformation measurements.

In this work, we first shortly summarize the fundamentals of InSAR technology. Next, we introduce the principles of the SD and MAI techniques, discussing their applications in real scenarios and addressing their theoretical and empirical accuracies. A significant concern is mainly on the estimation of the along-track (north–south) components of the ground deformations. Furthermore, we propose a new method for combining multi-track (multi-satellite) along-track ground displacement time-series, as obtained through advanced, multi-temporal MAI approaches (e.g., [88]), and recovering the combined time-series of the ground displacement along the north–south direction. To this end, we have adequately adapted the technique initially proposed in [75] to overcome the challenge of combing along-track MAI-driven measurements. Henceforth, by applying both the proposed method and original MinA technique, we have retrieved the north–south and the two-dimensional (up–down, east–west) ground displacement time-series of the investigated areas, thus recovering the (whole) 3-D dynamics of the observed deformation phenomena. Experimental results are presented by processing sequences of SAR data collected by the Envisat/ASAR, COSMO-SkyMed and Copernicus Sentinel-1 radar sensors relevant to the case studies of Afar Rift depression, Ethiopia and the 2019 Ridgecrest earthquakes, California. The results of some investigations proposed by other authors in recent years are also referenced throughout this manuscript.

The paper is organized as follows. Section 2 illustrates the InSAR rationale, emphasizing key strengths and potential limitations. Section 3 provides an overview of the SD and multi-aperture InSAR techniques for generating ground deformation maps related to single episodes as well as for the production of time-series of deformation by multi-temporal SAR acquisitions. In Section 4, we present a method for combining MAI-driven deformation measurements recovered from multiple data tracks to get enhanced N-S ground deformation time-series. Discussion and conclusions are finally addressed in Section 5.

## 2. Fundamentals of InSAR Technology and Applications

SAR Interferometry (InSAR) technology [14–16] plays a significant role in the context of microwave remote sensing. The InSAR methodology is applied to retrieve information on the state of the Earth's surface, its topography and its modifications. It relies on the extraction of the phase difference between two complex-valued SAR images related to the same imaged scene, usually referred to as master and slave, respectively. The multiplication between the master image and the complex conjugate of the slave image leads to the generation of a complex image (i.e., the complex interferogram) whose phase value is the phase difference between the two interfering SAR images. Such a complex interferogram takes into account the amplitude and phase returns of the observed target, and also contains a significant phase contribution due to the optical path from the target to the sensor at the collected times. Such a geometric phase contribution catches useful information on the geometry of the observed scene, such as

the topography of the imaged scene or modifications (i.e., the ground displacement) that have occurred along the sensor-to-target line-of-sight illumination. Depending on the relative positions of the sensing transmitting/receiving antennas, different InSAR investigations configurations arise (e.g., [7,89,90]):

(1) Across-track interferometry. In this case, the radar carrier has one or two sensors mounted on-board that are spaced along the across-track direction. Depending on the number of antennas on-board (one or two), the inferring SAR data pair can be acquired from different positions at different times (repeat pass interferometry), or from different locations at the same time (single-pass interferometry) (see Figure 1A,B). Across-track InSAR configuration is mainly used to map the Earth's crust and detect and monitor its changes [21,91–93].

(2) Along-track interferometry (ATI). In this case, two sensing antennas that are spaced in the along-track direction are mounted on-board the same carrier. Accordingly, two interfering SAR images can be acquired at different times from the same position (see Figure 1A,C). Along-track interferometry (ATI) configuration is usually used to estimate the radial motion of a surface point. In such a context, the ATI was firstly used for mapping the tidal ocean surface currents [94–97]. ATI is also principally applied for traffic monitoring with spaceborne SAR data [98–100].

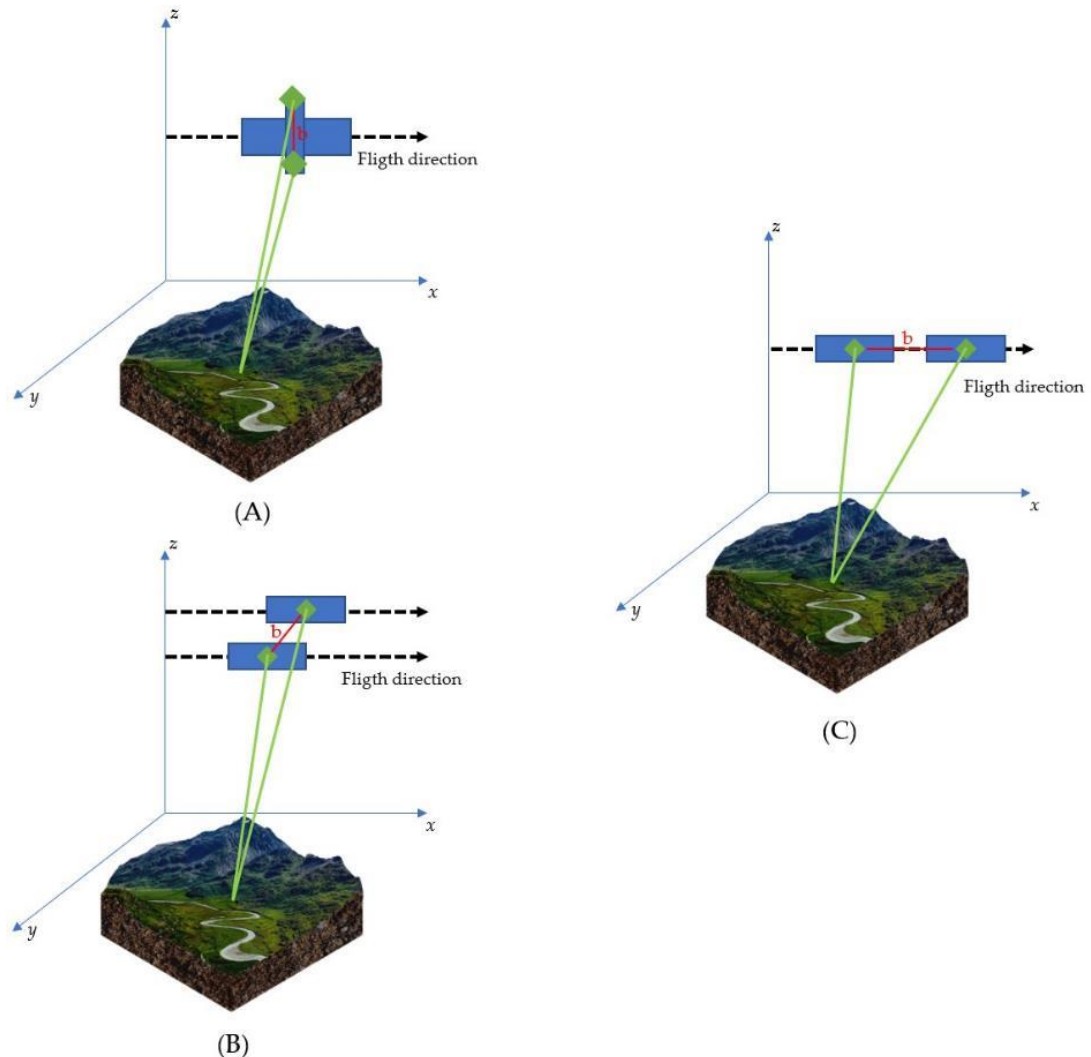

**Figure 1.** InSAR interferometry configurations: (**A**) single-pass across-track, (**B**) repeat-pass across-track and (**C**) along-track.

We review the basic principles of across track interferometry in the following subsections. Interested readers can find a more comprehensive overview of the InSAR techniques and their applications in the literature (e.g., [101,102]).

*InSAR for Topography Estimation*

The SAR interferometry technique exploits the difference of viewing angles in two (or more) acquisitions (e.g., across-track configuration) to estimate the topography of an observed scene (e.g., [9]). Considering repeat-pass across-track configuration (see Figure 1B), two interfering SAR images can be expressed in complex notation as:

$$I_1(x, r) = \gamma_1(x, r) e^{j\frac{4\pi}{\lambda}R} \tag{1a}$$

$$I_2(x, r) = \gamma_2(x, r) e^{j(\frac{4\pi}{\lambda}R + \delta_R)} \tag{1b}$$

where $\gamma_{1,2}(x, r)$ represents the complex-valued reflectivity function related to the target of azimuth and range coordinates $(x, r)$ and $j = \sqrt{-1}$ is the imaginary unit. Note that R and R + $\delta_R$ are the range distances between the sensor and the target in the two images, with $\delta_R$ being the additive range distance term due to slightly different viewing geometries. The phase difference of the two SAR images is obtained by extracting the phase of the complex interferogram [9], which is given by the complex conjugate product operation between the two SAR images, as follows:

$$I(x, y) = I_1(x, y) \cdot I_2^*(x, y) = \left| I_1(x, y) \cdot I_2^*(x, y) \right| exp \left[ j \frac{4\pi}{\lambda} \delta_R \right] \tag{2}$$

By taking into account the acquisition geometry shown in Figure 2 and applying the law of cosines, after trivial mathematical manipulations [103], it is straightforward to demonstrate that:

$$\Delta\phi = \frac{4\pi}{\lambda} \delta_R \cong \frac{4\pi}{\lambda} b \sin(\vartheta - \alpha) \tag{3}$$

where $\Delta\phi$ is the extracted interferometric phase. Note that $\lambda$ is the operational wavelength, $b$ is the interferometric baseline (i.e., the distance between orbital positions from which the two SAR images are taken), $\vartheta$ is the sensor side-looking angle and $\alpha$ is the angle the baseline makes with respect to a horizontal reference plane. Equation (3) contains useful information on the target height $h$. Indeed, by expanding Equation (3) around the angular position $\vartheta = \vartheta_0$, representing the illumination angle for flat terrain (i.e., $h$=0), we have:

$$\Delta\phi = \frac{4\pi}{\lambda} \cdot b[\sin(\vartheta_0 - \alpha) + \cos(\vartheta_0 - \alpha)(\vartheta - \vartheta_0)] \cong \Delta\phi_{flat} - \frac{4\pi}{\lambda} \cdot \frac{b_\perp}{R\sin(\vartheta)} h = \Delta\phi_{flat} - \Delta\phi_{topo} \tag{4}$$

where $\Delta\phi_{flat}$ is the so-called flat-Earth phase contribution, and $b_\perp$ is the component of the baseline that is perpendicular to the radar-to-sensor line-of-sight direction. The first term on the right-hand side of Equation (4), representing the flat-Earth phase term, can be derived (e.g., [93]) by computing the local phase frequency in the range direction as:

$$\Delta\phi_{flat} = -\frac{4\pi}{\lambda} \cdot \frac{b_\perp}{R\tan(\vartheta_0 - \Omega)} \rho \tag{5}$$

where $\Omega$ is the local slope of the terrain and $\rho = r$ is the pixel range coordinate of the imaged target. Conversely, the second phase term is the topographic phase signature $\Delta\phi_{topo}$. More general relations arise if the curvature of the Earth is also taken into account (e.g., [7]). If the range-dependent phase term is compensated for in Equation (4), the topography of the imaged scene $h$ is finally computed as [9]:

$$h = -\frac{\lambda R \sin\theta}{b_\perp} \cdot \frac{\Delta\varphi_{topo}}{4\pi} \tag{6}$$

From Equation (6), the height *h* can be estimated with an accuracy that is baseline-dependent; this means that the larger is the perpendicular baseline, the more accurate the estimation of topography will be [7]. However, very large baseline values are responsible for severe decorrelation noise artefacts [104]. They corrupt the measured interferometric phase $\Delta\phi$ and, subsequently, lead to incorrect height profile estimations. It is also worth highlighting that the interferometric phase is restricted to the [-π,π[ interval. The phase unwrapping (PhU) operation is a crucial step in any InSAR processing tool, and several approaches have been developed [105–108]. PhU involves the searching of (unknown) 2π-integer multiples that must be added to wrapped phases to compute unwrapped (full) phases.

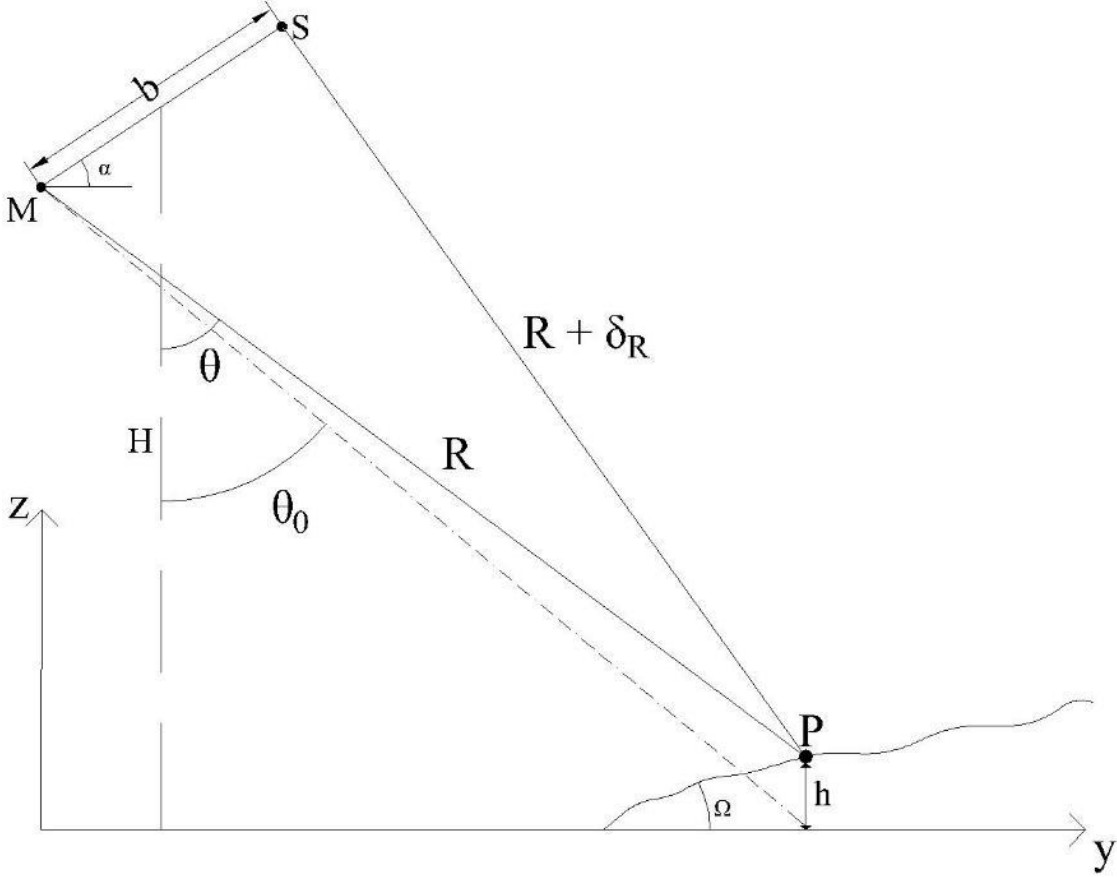

**Figure 2.** InSAR geometry acquisition. **M** and **S** represents the positions of master and slave acquisitions, respectively, and **P** is the generic point target on the ground.

As an evolution of the InSAR approach, a processing methodology known to as the differential synthetic aperture interferometric (DInSAR) technique [9,15,89] has been developed. It represents a common practice nowadays in the remote sensing scientific community for the detection and monitoring of the Earth's surface displacement phenomena. The key factor of the DInSAR technique with respect to other conventional approaches (e.g., GPS and levelling measurement campaigns) is that it allows the continuous monitoring of displacement phenomena with a dense grid of measurement points. To convey the Earth's surface displacements, the DInSAR technique relies on the generation of a so-called differential SAR interferogram [89]. Considering the acquisition geometry depicted in Figure 3, it is assumed that two complex-valued SAR images of the imaged scene were acquired at different times (see Figure 1B) and in different orbital positions. In a case where the ground surface is displaced by the 3-D vector **d** between the two flight passages of the sensor along the scene, the computed interferometric phase difference between the two SAR images—namely, $\Delta\phi$—is made up of two main phase components:

$$\Delta\phi = \Delta\phi' + \Delta\phi_{defo} = \Delta\phi' + \frac{4\pi}{\lambda}d_{LOS} \tag{7}$$

where $\Delta\phi'$ is the phase contribution in the absence of deformation, as expressed in Equation (4), and $\Delta\phi_{defo}$ is the additional phase contribution related to the ground displacement. Therefore, the phase signal associated with the deformation can be recovered by synthetically reconstructing the phase component contribution $\Delta\phi'$ and subtracting it, modulo $2\pi$, to the measured phase difference. In particular, $d_{LOS}$ is the projection of the displacement vector $d$ along the sensor-to-target LOS direction. The synthetic phase term $\Delta\phi'$ is simulated from the knowledge of external information on the acquisition geometry such as a digital elevation model (DEM) of the observed area, the orbit state-vectors and the operational parameters of the radar instrument [7,9]. In a more general case, the differential interferometric phase is expressed as:

$$\Delta\phi = \Delta\phi_{defo} + \Delta\phi_{topo} + \Delta\phi_{orb} + \Delta\phi_{atmo} + \Delta\phi_{noise} \tag{8}$$

where $\Delta\phi_{topo} = \frac{4\pi}{\lambda} \cdot \frac{b_\perp}{r\sin\theta}\Delta z$ represents the residual topography phase component induced by the DEM errors $\Delta z$; $\Delta\phi_{orb}$ represents the residual phase term related to inaccurate orbital parameter information, which is used to estimate the aforementioned synthetic phase term; $\Delta\phi_{atmo}$ denotes the phase components relative to the propagation variation of the RADAR signal caused by the absorption effects of Earth's atmosphere; and, finally, $\Delta\phi_{noise}$ is representative of additive noise contributions (i.e., spatial and temporal decorrelation, incorrect focusing of SAR raw data, etc.), that corrupt the interferometric phase [104,109]. In particular, the spatial decorrelation is less pronounced for short perpendicular baseline configurations.

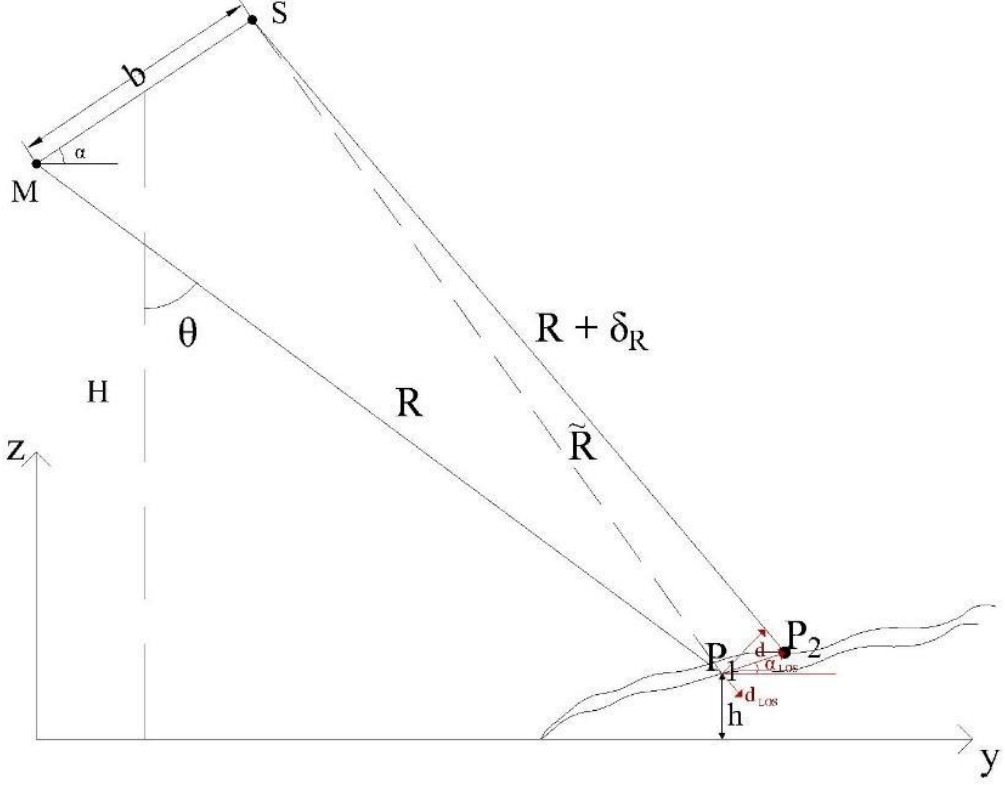

**Figure 3.** DInSAR geometry acquisition. Same as Figure 2 but considering the displacement **d** of the imaged point **P** on the ground. Note that **P₁** and **P₂** represent the point target positions of the master and slave acquisitions, respectively.

From Equation (7), the accuracy of the displacement measurements depends on the operational wavelength $\lambda$. In this case, the maximum detectable unambiguous displacement that corresponds to a full phase cycle of $2\pi$ is equal to $\frac{\lambda}{2}$, denoting measurement accuracy of a fraction of the employed wavelength that is dependent on the amount of noise present in the measured phase.

Interested readers can find an overview of DInSAR and its main applications in [93,110].

## 3. Spectral Diversity and Multiple Aperture Interferometry

In this section we present spectral diversity approaches that can be exploited by the multiple aperture interferometry methods and used for the computation of large deformation phenomena of the solid Earth, as well as for correction of the co-registration errors of TOPS mode Sentinel-1 SAR data.

### 3.1. Spectral Diversity

Spectral diversity (SD) techniques [111,112] represent a class of methods based on the calculations of the spectral properties of pairs of complex-valued SAR images. The basic method consists of calculating the spectral separation of sub-bands of SAR images obtained by cutting two different frequency slices along the azimuth (Doppler band) of range directions, or by splitting the azimuth (range) spectrum of a full-band SAR image using a proper band-pass filter. Let us describe the SD process by initially referring to the azimuth case, where we assume the availability of two full-band Single Look Complex (SLC) images, referred to as the master and slave images $I_1(x, r)$ and $I_2(x, r)$, respectively. If the two SLC images are focused (squinted geometry) with the same Doppler centroid (DC) frequency—namely, $f_{DC}$—the two SAR images can be expressed after the co-registration step [85] as:

$$I_M(x, r) = \gamma_M(x, r) \cdot exp\left[j\frac{4\pi}{c}f_c \cdot r\right] \cdot exp\left[j2\pi\frac{f_{DC}}{v} \cdot x\right] \tag{9a}$$

$$I_S(x, r) = \gamma_S(x, r) \cdot exp\left[j\frac{4\pi}{c}f_c \cdot (r + \delta r)\right] \cdot exp\left[j2\pi\frac{f_{DC}}{v} \cdot (x + \delta x)\right] \tag{9b}$$

where $f_c$ is the radar carrier frequency, $v$ is the platform velocity and $\delta x$ represents the azimuthal misregistration of the slave image with respect to the reference master image. Accordingly, an additional interferometric phase term is present. Indeed,

$$\Delta\phi = \frac{4\pi}{\lambda}d_{LOS} + 2\pi\frac{f_{DC}}{v}\delta x \tag{10}$$

Consequently, if two azimuth sub-bands are centered on low and high DC frequencies ($f_{DC}^-$ and $f_{DC}^+$, respectively (see Figure 4), four SAR images can be formed:

$$I_M^-(x, r) = \gamma_M(x, r) \cdot exp\left[j\frac{4\pi}{c}f_c \cdot r\right] \cdot exp\left[j2\pi\frac{f_{DC}^-}{v} \cdot x\right] \tag{11a}$$

$$I_S^-(x, r) = \gamma_S(x, r) \cdot exp\left[j\frac{4\pi}{c}f_c \cdot (r + \delta r)\right] \cdot exp\left[j2\pi\frac{f_{DC}^-}{v} \cdot (x + \delta x)\right] \tag{11b}$$

$$I_M^+(x, r) = \gamma_M(x, r) \cdot exp\left[j\frac{4\pi}{c}f_c \cdot r\right] \cdot exp\left[j2\pi\frac{f_{DC}^+}{v} \cdot x\right] \tag{11c}$$

$$I_S^+(x, r) = \gamma_S(x, r) \cdot exp\left[j\frac{4\pi}{c}f_c \cdot (r + \delta r)\right] \cdot exp\left[j2\pi\frac{f_{DC}^+}{v} \cdot (x + \delta x)\right] \tag{11d}$$

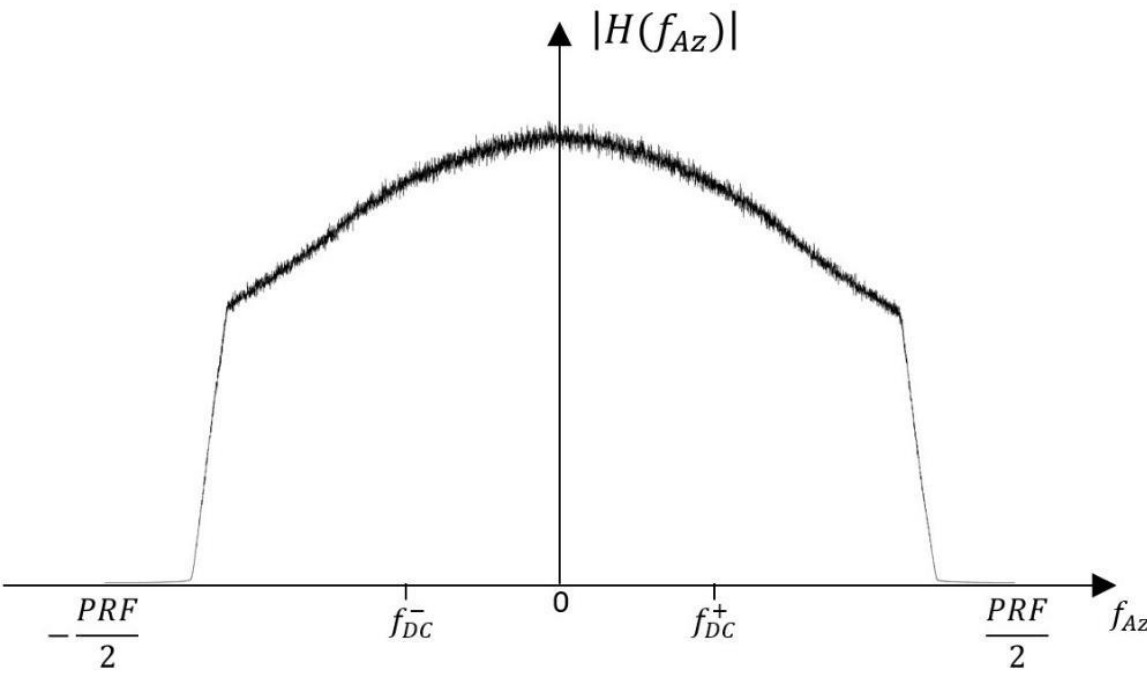

**Figure 4.** Sketch of the amplitude azimuth spectrum ($H_{fAz}$) of a full-band SAR image, where $f_{DC}^-$ and $f_{DC}^+$ indicate the low and the high Doppler centroid (DC) frequencies, respectively, and PRF indicates the sensor's pulse repetition frequency.

This means that, for each image of the SAR data pair, two images are determined that are related to the low and high components of SAR data pair spectra. Subsequently, from these two data pairs, two interferograms are generated. After the split-spectrum operation, the interferometric phase terms related to the lower and higher azimuthal sub-bands are expressed as follows:

$$\Delta\phi^- = \frac{4\pi}{\lambda}d_{LOS} + 2\pi\frac{f_{DC}^-}{v}\delta x \tag{12a}$$

$$\Delta\phi^+ = \frac{4\pi}{\lambda}d_{LOS} + 2\pi\frac{f_{DC}^+}{v}\delta x \tag{12b}$$

Therefore, the wrapped phase difference between the lower and higher spectral bands is

$$\widetilde{\Delta\phi} = Wr\left(\Delta\phi^- - \Delta\phi^+\right) = 2\pi\frac{f_{DC}^- - f_{DC}^+}{v}\delta x \tag{13}$$

where $Wr(\cdot)$ is the wrapping operator that wraps out the phase in the $[-\pi, \pi[$ interval. As a consequence, the azimuth misalignment $\delta x$ can be computed from Equation (13). Assuming that such a misalignment is small enough to avoid the measured phase becoming ambiguous, the term $\delta x$ can easily be derived as

$$\delta x = \frac{v\widetilde{\Delta\phi}}{2\pi\left(f_{DC}^- - f_{DC}^+\right)} \tag{14}$$

From Equation (14), it is evident that the larger the spectral separation between the central frequencies of the two sub-bands, the more accurate the estimation of $\delta x$ is. We would like to stress that in non-stationary scenarios, the imaged target at azimuth location $x$ is subject to displacement, —namely, $\delta x_{disp}$—and, in this case, the measured $\delta x$ term contains both a contribution due to the azimuthal misregistration $\delta x_{misr}$ and the azimuthal $\delta x_{disp}$ displacement. Of course, misalignments of SAR data pairs can be thoroughly controlled in the imaging process to mitigate phase decorrelation

effects. Perfectly aligned SAR data are then used to infer geophysical displacement measurements, as they have been via the multiple aperture SAR interferometry (MAI) technique in cases of large ruptures in the ground [41,113].

We would like to remark that SD and splitting-band methods can also be applied along the range direction by extracting the two sub-bands in the range. In particular, we begin with two range sub-bands centered on the two frequencies $f_c^-$ and $f_c^+$ (see Figure 5), which can be expressed, similarly to Equations (11a-d), as

$$I_M^-(x,r) = \gamma_M(x,r) \cdot exp\left[j\frac{4\pi}{c}f_c^- \cdot r\right] \cdot exp\left[j2\pi\frac{f_{DC}}{v} \cdot x\right] \tag{15a}$$

$$I_M^-(x,r) = \gamma_M(x,r) \cdot exp\left[j\frac{4\pi}{c}f_c^- \cdot r\right] \cdot exp\left[j2\pi\frac{f_{DC}}{v} \cdot x\right] \tag{15b}$$

$$I_M^+(x,r) = \gamma_M(x,r) \cdot exp\left[j\frac{4\pi}{c}f_c^+ \cdot r\right] \cdot exp\left[j2\pi\frac{f_{DC}}{v} \cdot x\right] \tag{15c}$$

$$I_S^+(x,r) = \gamma_S(x,r) \cdot exp\left[j\frac{4\pi}{c}f_c^+ \cdot (r+\delta r)\right] \cdot exp\left[j2\pi\frac{f_{DC}}{v} \cdot (x+\delta x)\right] \tag{15d}$$

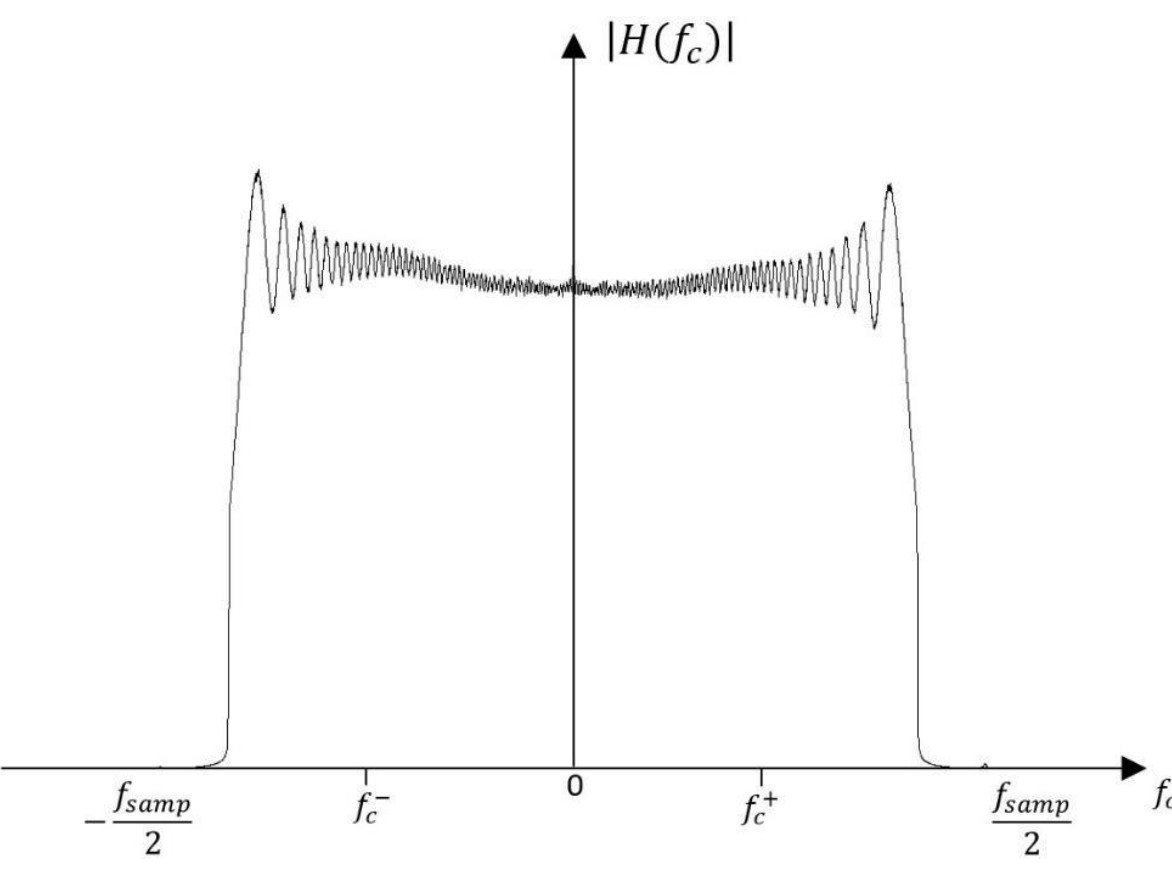

**Figure 5.** Sketch of the amplitude range spectrum ($H_{fc}$) of a full-band SAR image, where $f_c^-$ and $f_c^+$ indicate the low and the high range frequencies, respectively, and $f_{samp}$ is the sampling frequency.

In this case,

$$\Delta\phi^- = \frac{4\pi}{c}f_c^- \cdot d_{LOS} + 2\pi\frac{f_{DC}}{v}\delta x + \Delta\phi_n^- \tag{16a}$$

$$\Delta\phi^+ = \frac{4\pi}{c}f_c^+ \cdot d_{LOS} + 2\pi\frac{f_{DC}}{v}\delta x + \Delta\phi_n^+ \tag{16b}$$

where $\Delta\phi_n^-$ and $\Delta\phi_n^+$ are the negative and positive sub-band phase noise contributions, respectively. Accordingly, the wrapped phase difference of the higher- and lower-band phase terms gets an alternative estimate of the LOS displacement as

$$d_{LOS} = \frac{c\Delta\widetilde{\phi}}{4\pi\left(f_c^- - f_c^+\right)} \tag{17}$$

where, similarly to Equation (13), $\Delta\widetilde{\phi}$ represents the wrapped phase difference $Wr(\Delta\phi^- - \Delta\phi^+)$ between the two phase looks (see Equation (16b)). Note that $d_{LOS}$ is assumed to be small enough to avoid phase measurements ambiguities. In this framework, a procedure that is known as spectral shift filtering [114] implements proper filters that are tuned on specifically chosen different central frequencies to compensate for the geometrical decorrelation effects that corrupt the interferometric phase. This strategy has been applied correctly to process coupled SAR images collected by the ERS and the Envisat systems, which were characterized by a spectral shift of 31 MHz [115,116].

As final remark, we would like to stress that only a fraction $\beta$ of the total bandwidth of the two images is used after the application of the split-spectrum operation. Accordingly, being the spatial resolution inversely proportional to the band of the signal [89], the spatial resolution of the two sub-band interferograms is reduced by the same fraction.

### 3.2. Multiple Aperture Interferometry Principles

As said in Section 2, InSAR techniques have successfully been used to study and analyze many different phenomena that characterize the Earth's surface. However, these techniques are limited to measuring only the 1-D components of ground deformation along the RADAR LOS direction. Conversely, enabling 3-D displacement measurements is crucial for better analyses and to study the deformation phenomena that characterize areas of interest. In the last two decades, many studies have been carried out enabling the measurement of the 3-D (up–down (U-D), east–west (E-W) and north–south (N-S)) ground displacement [66,67,117]. Multi-platform, multi-track InSAR techniques have been developed to discriminate the U-D and E-W components of surface displacement, but estimation of N-S ground displacement data has been more challenging. In this respect, SAR amplitude tracking methods—namely, pixel offset (PO) [32,42,48,65,68,118] and multiple aperture interferometry (MAI) [41,49,52,53]—have been applied.

It has been demonstrated that the measurement accuracy of N-S displacement that is attainable with PO methods is low. In particular, it reaches only a fraction of pixel spacing (1/30th), as the azimuthal pixel spacing of first-generation and present-day SAR instruments is in the order of 3–5 cm [32,44–46,48]. For example, taking into account that the azimuth pixel spacing of the TerraSAR-X sensor is of about 3 m, the expected accuracy of PO measurements is about 10 cm. Of course, fine and ultra-fine SAR data with enhanced spatial resolutions are envisaged to apply the PO method. It is worth noting that a substantial improvement in the discrimination of the N-S ground displacement component from InSAR data is a key factor in performing extended geophysical investigations.

### 3.3. Multiple Aperture Interferometry for the Along-Track Measurement

Monitoring the Earth's surface phenomena requires more comprehensive analyses, such as estimating the 3-D field of ground surface displacement that affects a specific area. Since SAR satellite platforms travel in almost near-polar orbits, the main problem in the discrimination of the north–south component of deformation (approximately along-track displacement) is that the projection of the ground displacement along the north–south direction is not accurate enough [41].

In recent years, a big improvement in the discrimination of along-track displacement has been through development of the MAI technique. This technique uses a split beam process called "spectral diversity" (see Section 3.1) to determine backward- and forward-looking interferograms [111] from a SAR data pair that is related to the same scene but were acquired at different times (see Figure 6).

Considering Equation (13), the backward- and forward-looking interferograms can be expressed as follows:

$$\phi_{fw} = \frac{2\pi}{v} \cdot f_{DC}^{+} x \tag{18a}$$

$$\phi_{bw} = \frac{2\pi}{v} \cdot f_{DC}^{-} x \tag{18b}$$

Specifically,

$$f_{DC}^{+} = f_{DC,c} + n \cdot \frac{\Delta f_D}{2} \tag{19a}$$

$$f_{DC}^{-} = f_{DC,c} - n \cdot \frac{\Delta f_D}{2} \tag{19b}$$

where $f_{DC,c} = \frac{f_{DC,m} + f_{DC,s}}{2}$ and $\Delta f_D = \frac{2v}{l}$ are the average Doppler centroid (DC) frequencies of the two full-bandwidth SAR images, with $f_{DC,m}$ and $f_{DC,s}$ being the DCs of the master and slave SAR images, respectively. Note that $\Delta f_D$ is the effective Doppler bandwidth, $l$ indicates the azimuthal antenna length (see Figure 7A) and $n$ is the fraction of the azimuth bandwidth (e.g., $\Delta f_D$; see Figure 7A) used in the split-beam process, also indicated as the normalized squint angle of the sub-aperture process. Consequently, starting from the measured phase difference between the forward- and backward-looking interferometric phases, the MAI phase is determined as follows:

$$\phi_{MAI} = Wr(\phi_{fw} - \phi_{bw}) = \frac{2\pi}{v} n \cdot \Delta f_D \cdot \Delta x = \frac{2\pi}{v} n \left( \frac{2v}{l} \right) \Delta x = \frac{4\pi}{l} n \Delta x + \eta \tag{20}$$

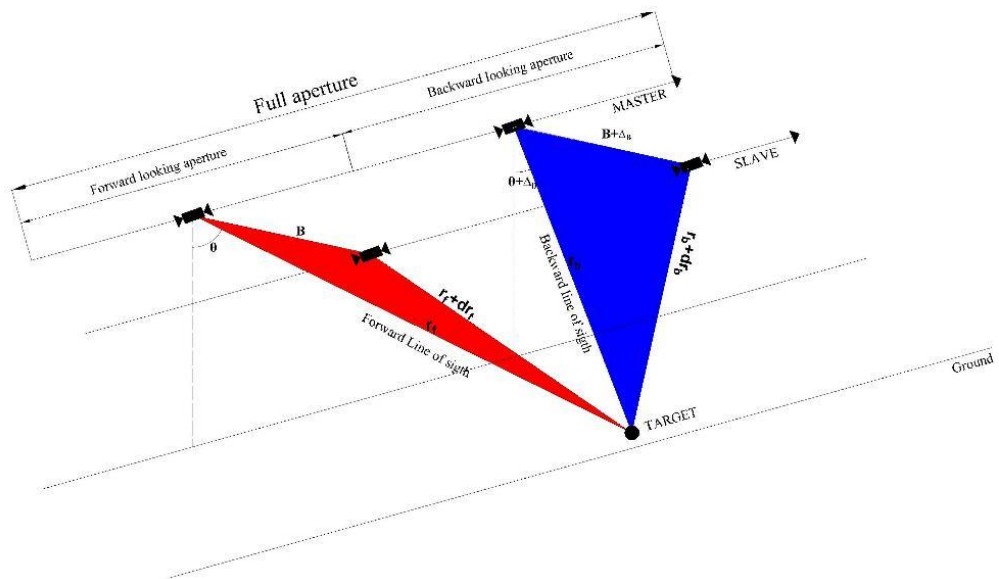

**Figure 6.** Multiple aperture interferometry geometry acquisition. The target on the ground is observed in the master and slave acquisitions by the SAR sensor with slightly different viewing geometries. Forward- and backward-looking apertures are produced by the split beam process.

In Equation (20), it is clear that the MAI phase is proportional to the along-track deformation $\Delta x$, where, for the sake of simplicity, we have imposed $f_{DC,c} = 0$, and $\eta$ is the interferometric noise. Naturally, Equation (20) represents the simplest case. In particular, during the MAI interferogram determination, the SD process generates the forward- and backward-looking interferometric phases, respectively, splitting the full azimuthal spectral bandwidth of the master and slave images (see Figure 7B).

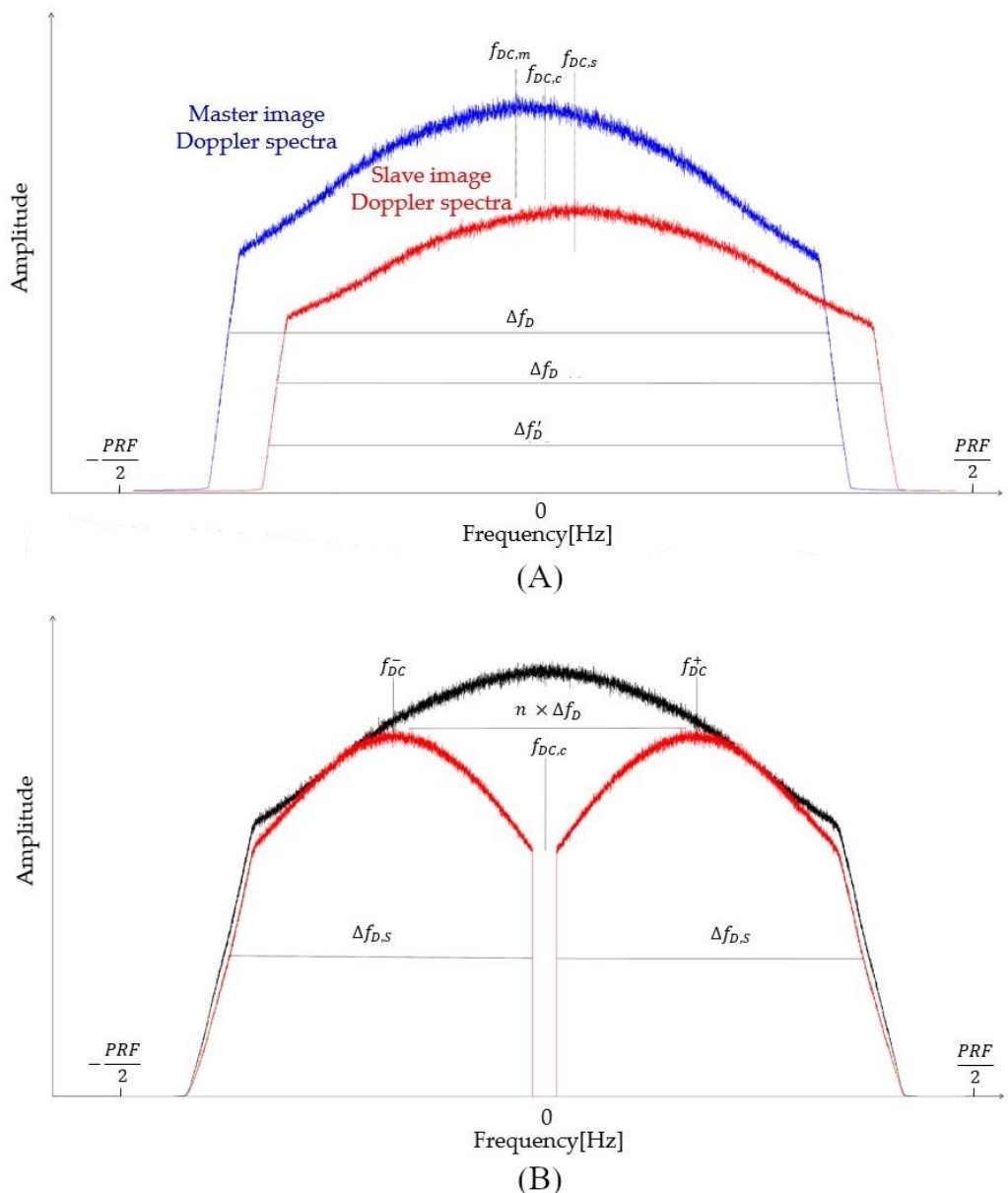

**Figure 7.** (**A**) Doppler frequency spectra of a pair of Envisat/ASAR complex images related to the Afar depression acquired on 19 December 2005 (blue spectra) and on 25 August 2008 (red spectra). (**B**) Doppler frequency spectra filtered using two Hamming windows (red colored spectra).

If the master and slave acquisition orbits are not perfectly parallel when considering the backward- and forward-looking imaging geometries (see Figure 6), different flat-Earth and topographic phase contributions arise in the forward- and backward-looking interferograms. Considering Equations (4) and (5), Jung et al. [113] addressed this problem by considering the expression of the additional spurious phase terms related to the flat-Earth and the topographic contributions, which are related to the baseline difference between the forward- and backward-looking imaging geometries, namely, $\Delta B_\perp$:

$$\phi_{flat} = \frac{4\pi\Delta B_\perp}{\lambda R\tan(\theta)}\rho \tag{21a}$$

$$\phi_{topo} = \frac{4\pi\Delta B_\perp}{\lambda R\sin(\theta)}h \tag{21b}$$

where $\rho$ and $h$ are the slant range and topographic height, respectively. The atmospheric phase screen (APS) may be considered negligible in the final MAI phase, as APS is influential in the same way both the backward- and forward-looking interferograms are.

In this context, Jung et al. [113] presented a MAI improvement process on which they estimated and corrected for the mentioned flat-Earth and topographic spurious phase terms. Afterwards, we will present two case studies related to the application of the MAI technique to the investigation of a single deformation episode.

The first case study concerns the Afar depression. Afar is a region situated in Ethiopia, Africa, that was affected in 2005 by a big earthquake associated with the Ethiopian Great Rift fault mechanism [119,120].

Specifically, from September to October 2005, a seismic sequence consisting of 131 events affected the Afar region as a result of the Dabbahu volcano eruption (see Figure 8A). Since the initial main events of the drifting episode, a further sequence of 13 discrete dyke events was detected from 2005 to 2010 along the entire Dabbahu rift segment [119].

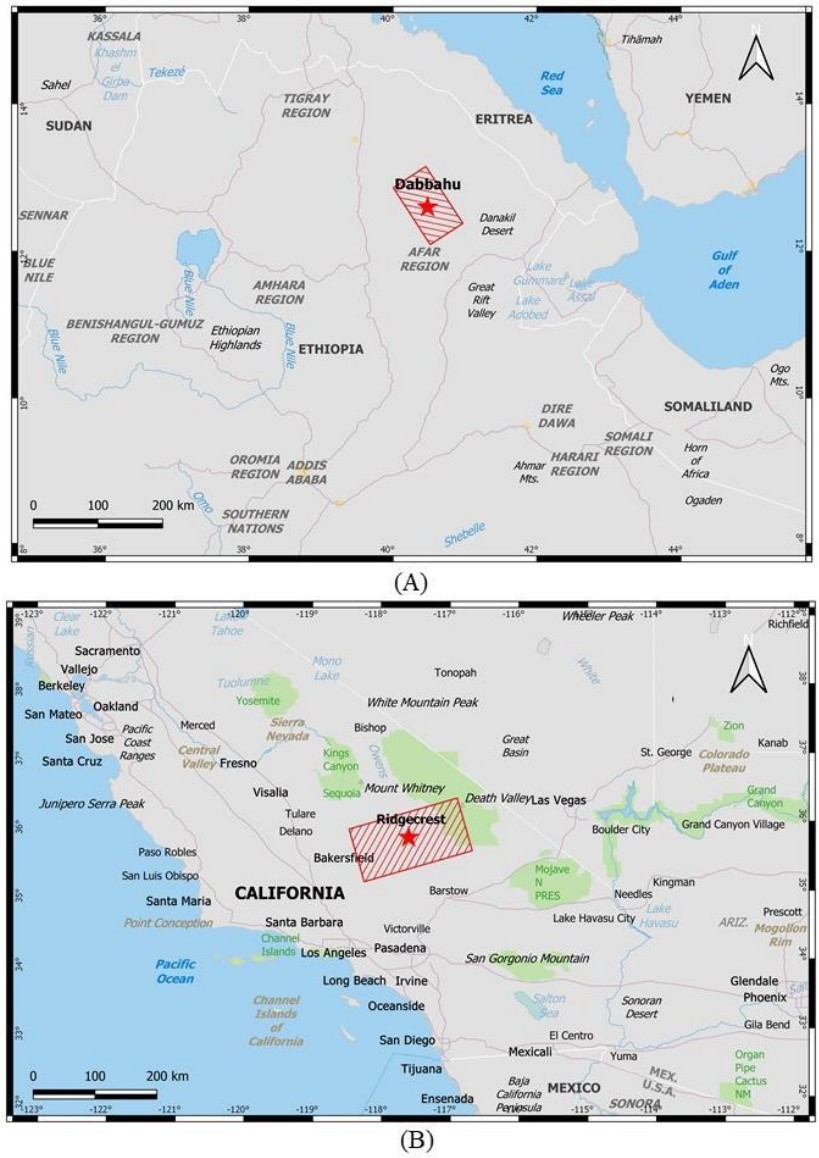

(A)

(B)

**Figure 8.** Geographical maps of the case study areas. (**A**) Afar depression, Ethiopia, and (**B**) Ridgecrest, California, USA. Red rectangles in (**A**,**B**) represent the footprints of the used ASAR/Envisat and Sentinel-1 datasets, respectively.

We performed an experiment by considering one pair of Envisat/ASAR data that captured the effects of the Dabbahu rifting episode. SAR data were acquired along ascending orbits (Track 200) on 19 December 2005 (master) and 25 August 2008 (slave), as shown in Figure 9A,B. Starting from this pair of SAR data, a MAI interferogram was generated (see Figure 10A). The forward- and backward-looking interferograms of the Dabbahu region area are shown in Figure 10B,C. In particular, we have highlighted an area in the MAI interferogram with a red circle that showed sensitive ground displacement and was in correspondence with a trace of the 2005 activated rift segment.

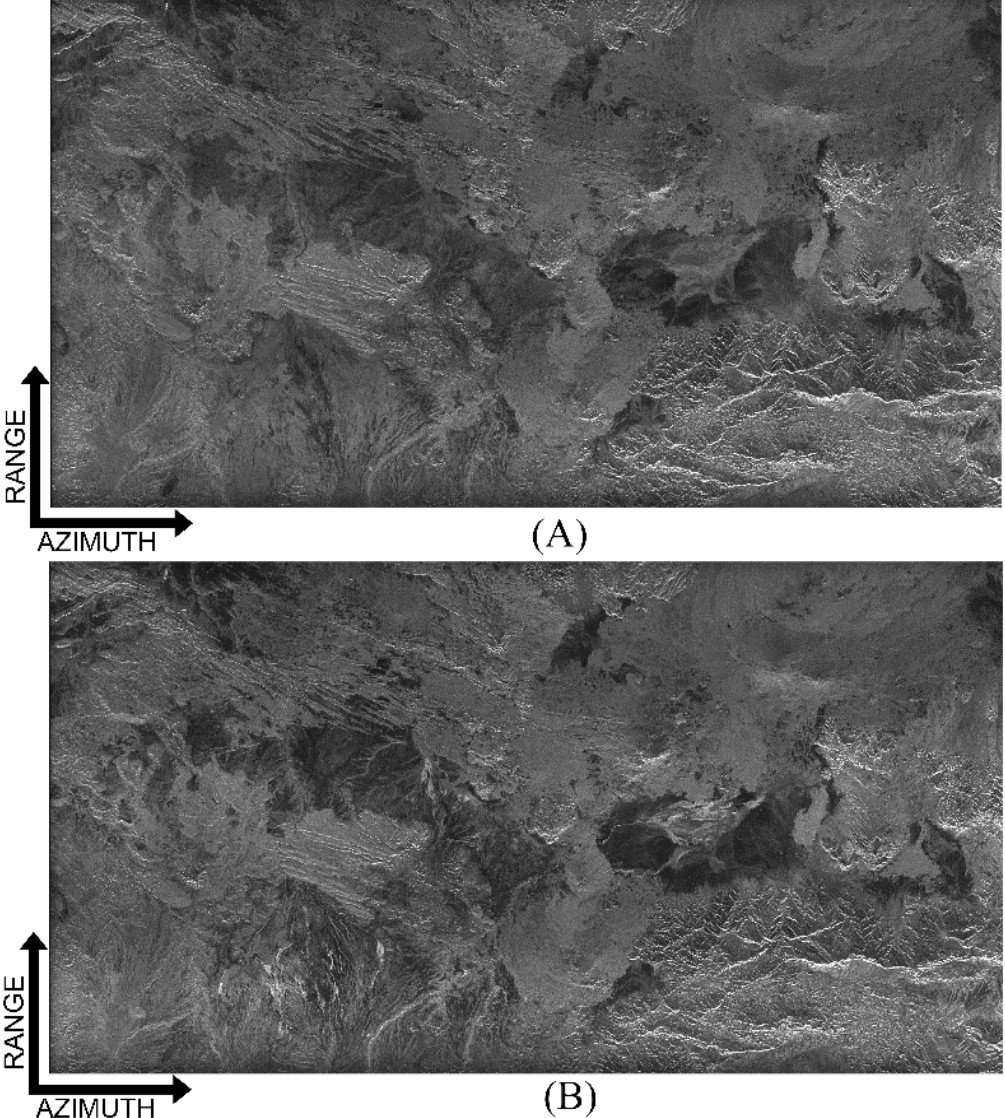

**Figure 9.** Afar depression region. Amplitude SAR images of the ASAR master (**A**) and slave (**B**) acquisitions, collected on 19 December 2005 and 25 August 2008, respectively.

The second case study was performed by processing a couple of SAR data acquired over the north-east Ridgecrest town area, situated in California, USA, (see Figure 8B), which was struck by a big (Mw) 7.1 earthquake on 6 July 2019, the strongest in the region in at least two decades. The phenomena included other previous main shock events, and in the days and weeks that followed, thousands of aftershocks rumbled beneath Southern California [121].

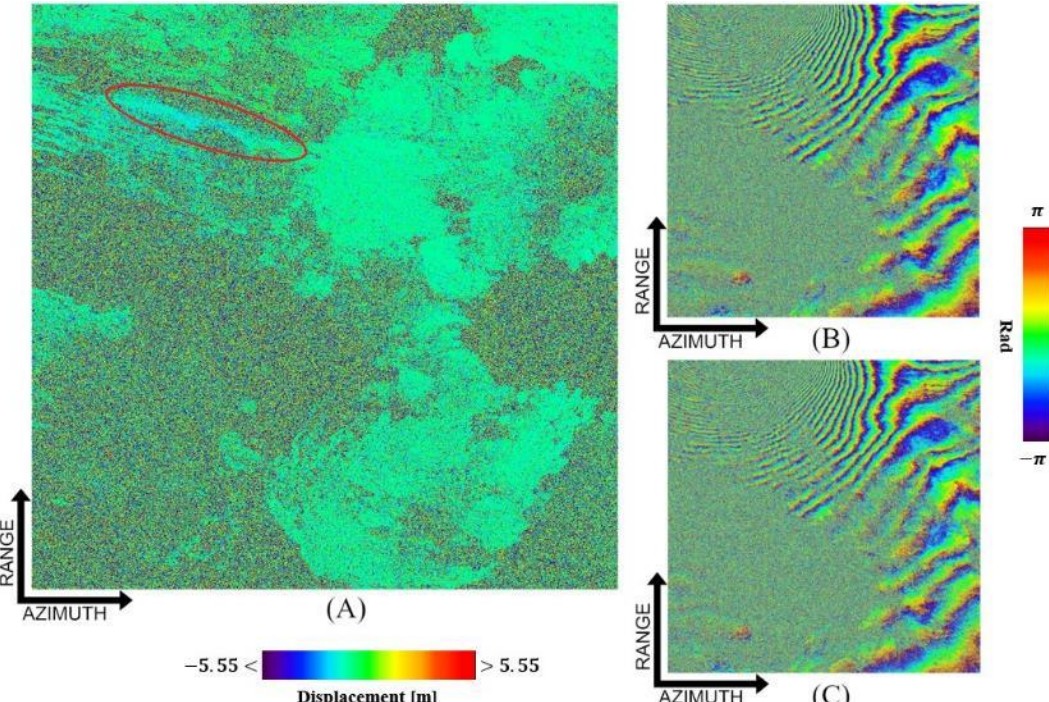

**Figure 10.** (**A**) Multiple aperture SAR interferometry (MAI) and (**B**,**C**) forward- and backward-looking interferograms of the Afar depression zone, generated from a pair of the ASAR acquisitions that were collected on 19 December 2005 and 25 August 2008, respectively. Red circle identifies the main fault trace.

In particular, starting from a single pair of COSMO-SkyMed acquisitions—the master collected on 4 July 2019, and the slave on 20 July 2019 (see Figure 11A,B)—a MAI interferogram (Figure 12A) was generated from a pair of forward- and backward-looking interferograms (Figure 12B,C). Specifically, in the interferogram depicted in false colors, we can clearly see the big fault related to the main seismic event circled in red (despite the presence of the interferometric noise) at the center.

Considering the parameters of the X-band COSMO-SkyMed RADAR instrument of the Italian Space Agency (ASI) and the Envisat/ASAR of European Space Agency (ESA)—with azimuthal antenna lengths of 5.7 and 11.1 m, respectively—we evaluated the along-track measurement enabled by the MAI technique. In particular, taking into account Equation (20) and neglecting the noise term (for the sake of simplicity), we observe that an ambiguous maximum along-track deformation $\Delta x$ of about 5.7 and 11.1 m for every in $2\pi$ phase cycle can be measured. This result implicitly confirms the useful utility of the MAI technique for estimating and analyzing large displacements of the Earth's surface. A sounder analysis of MAI measurement performance is presented in Section 3.4. The processing scheme of MAI operations is shown in Figure 13.

### 3.4. Multiple Aperture Interferometry Accuracy and Noise propagation

Based on Equation (20), the accuracy of the MAI technique can be theoretically estimated by the following Equation (22) that relates the standard deviation of the MAI phase measurement $\sigma_{\varphi_{MAI}}$, with the standard deviation of the along-track deformation measurements $\sigma_x$ [113]:

$$\sigma_x = \frac{l}{4\pi \cdot n}\sigma_{\varphi_{MAI}} \tag{22}$$

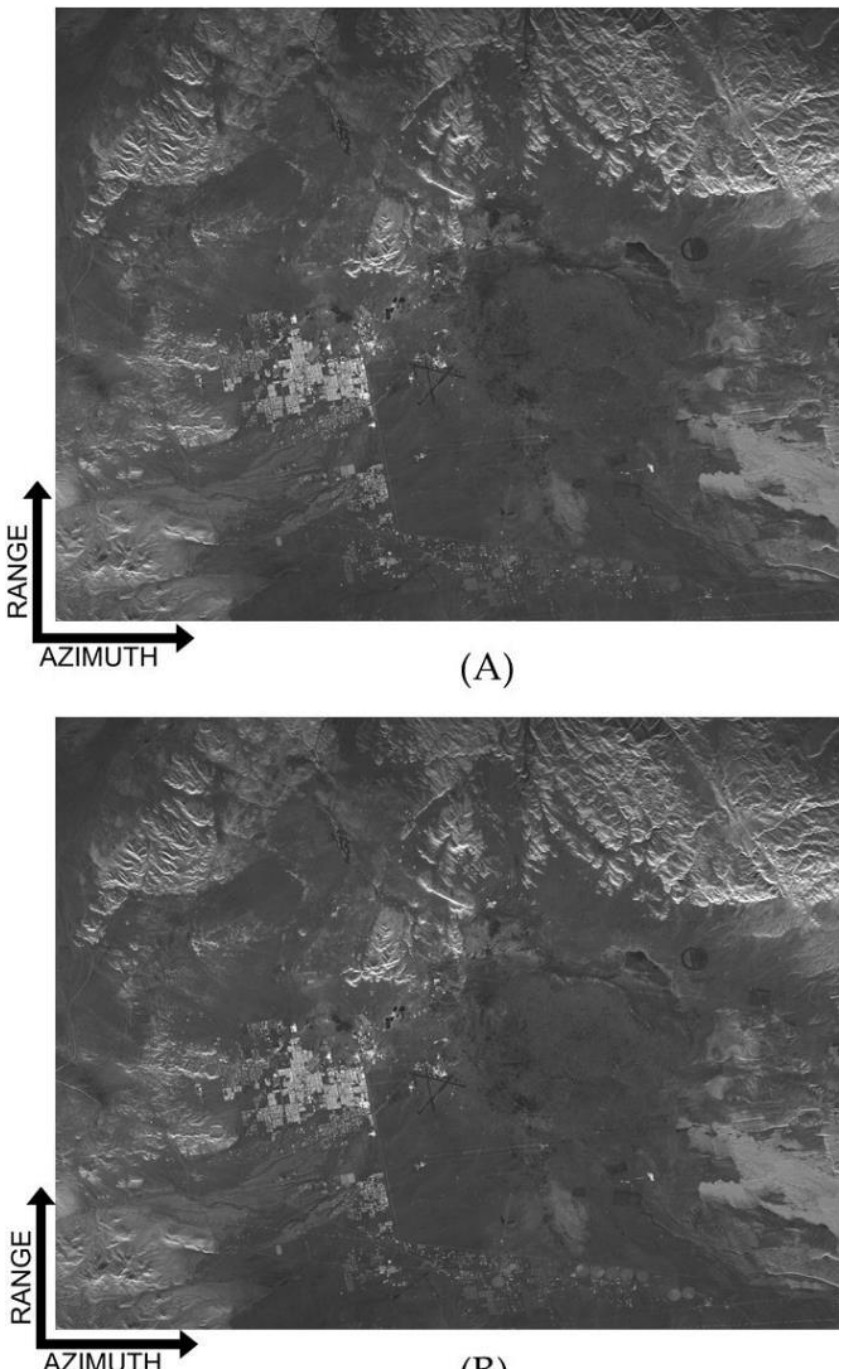

**Figure 11.** Area of Ridgecrest, California, USA. Amplitude SAR images of the COSMO-SkyMed master (**A**) and slave (**B**) acquisitions, acquired on 4 July 2019 and 20 July 2019, respectively.

From Equation (22), it is clear that the MAI measurement accuracy strictly depends on the standard deviation of the interferometric phase noise [112], which is given by

$$\sigma_{\varphi_{MAI}} = \sqrt{\sigma_{fw}^2 + \sigma_{bw}^2 - 2\sigma_{fw,bw}} \qquad (23)$$

where $\sigma_{fw}^2, \sigma_{bw}^2, \sigma_{fw,bw}$ are the variances of forward- and backward-looking interferograms and their co-variance, respectively. Thus, based on the assumption that the probability density function (pdf) of the single/multi-look interferometric phases can be expressed as a function of the correlation

coefficient $\gamma$ [122], and assuming the forward- and backward-looking interferograms are two statistically independent random variables (e.g., $\sigma_{\varphi,fb}$ is zero) in the limit of Cramér–Rao [112] (which is valid for high values of $\gamma$), $\sigma_{\varphi,f}$ and $\sigma_{\varphi,b}$ can be inferred as follows:

$$\sigma_{fw} = \sigma_{bw} \approx \frac{1}{\sqrt{2NL}} \cdot \frac{\sqrt{1-\gamma^2}}{\gamma} \tag{24}$$

where $NL$ is the effective number of looks. Finally, using Equations (23) and (24), the standard deviation of the interferometric MAI phase is reduced to

$$\sigma_{\varphi_{MAI}} \approx \frac{1}{\sqrt{NL}} \cdot \frac{\sqrt{1-\gamma^2}}{\gamma} \tag{25}$$

where we assume that the forward- and backward-looking interferograms are uncorrelated. From Equations (22) and (25), it is obvious that improving the coherence of the MAI interferogram and having a larger spectral separation between the sub-band both lead to better accuracy in the estimation of along-track deformation measurements. For the Cramér–Rao boundary, it was demonstrated in [77,112] that the optimal separation between the sub-bands is equal to two-thirds of the full bandwidth, and that:

$$\sigma_{\varphi_{MAI,CR}} \approx \sqrt{\frac{3}{NL}} \cdot \frac{\sqrt{1-\gamma^2}}{\gamma} \tag{26}$$

By considering Equation (26), we can plot the standard deviation of the azimuthal displacement vs the spatial coherence of the MAI interferogram (see Figure 14A,B) for different values of the effective look numbers ($NL$), which comprise the operational parameters of the Envisat/ASAR, COSMO-SkyMed and Sentinel-1 platforms listed in Tables 1–3, respectively. In particular, as outlined in Section 3.6, the Doppler bandwidth of the TOPS-mode SAR data is reduced for a single point target with respect to the conventional stripmap case by a factor of about three (see Equation (20)).

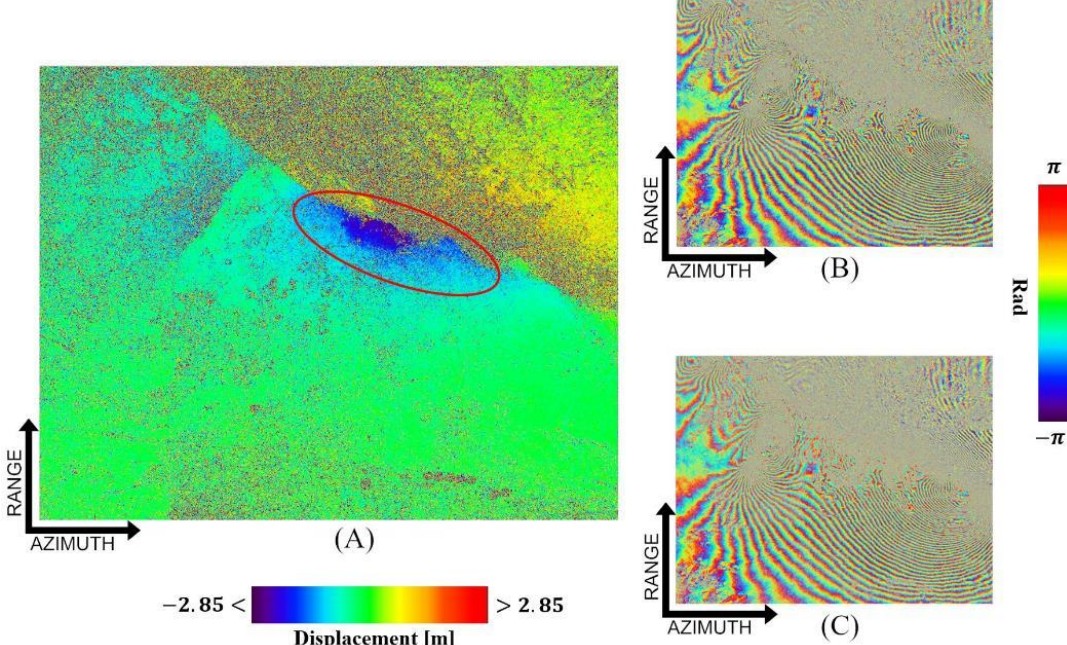

**Figure 12.** (**A**) MAI and (**B**,**C**) forward- and backward-looking interferograms of the area of Ridgecrest, California, USA, generated from a pair of COSMO-SkyMed acquisitions collected on 4 July 2019 and 20 July 2019, respectively. The red circles identify the fault line zone.

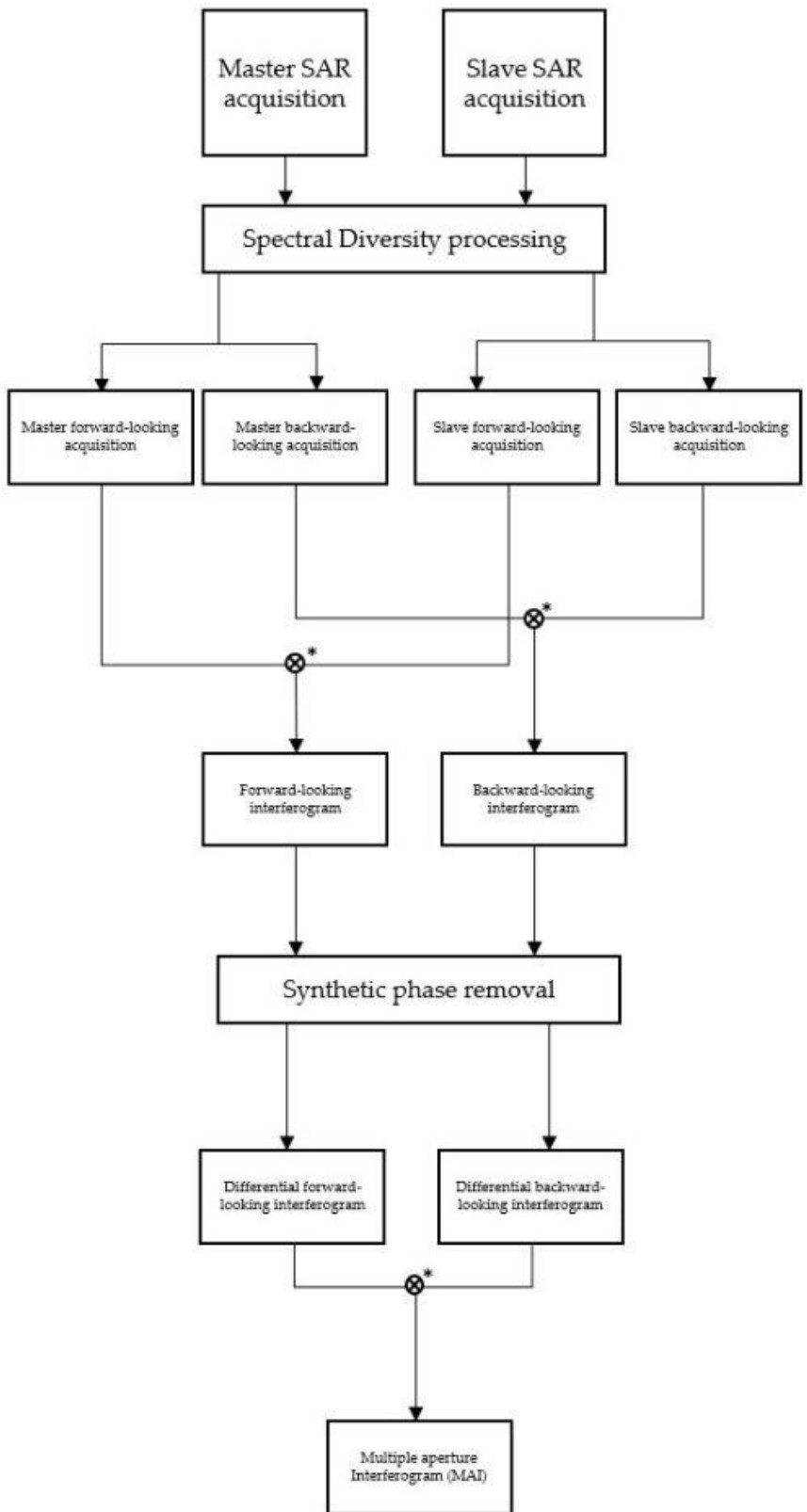

**Figure 13.** MAI processing flowchart. The symbol * denotes a complex conjugate multiplication.

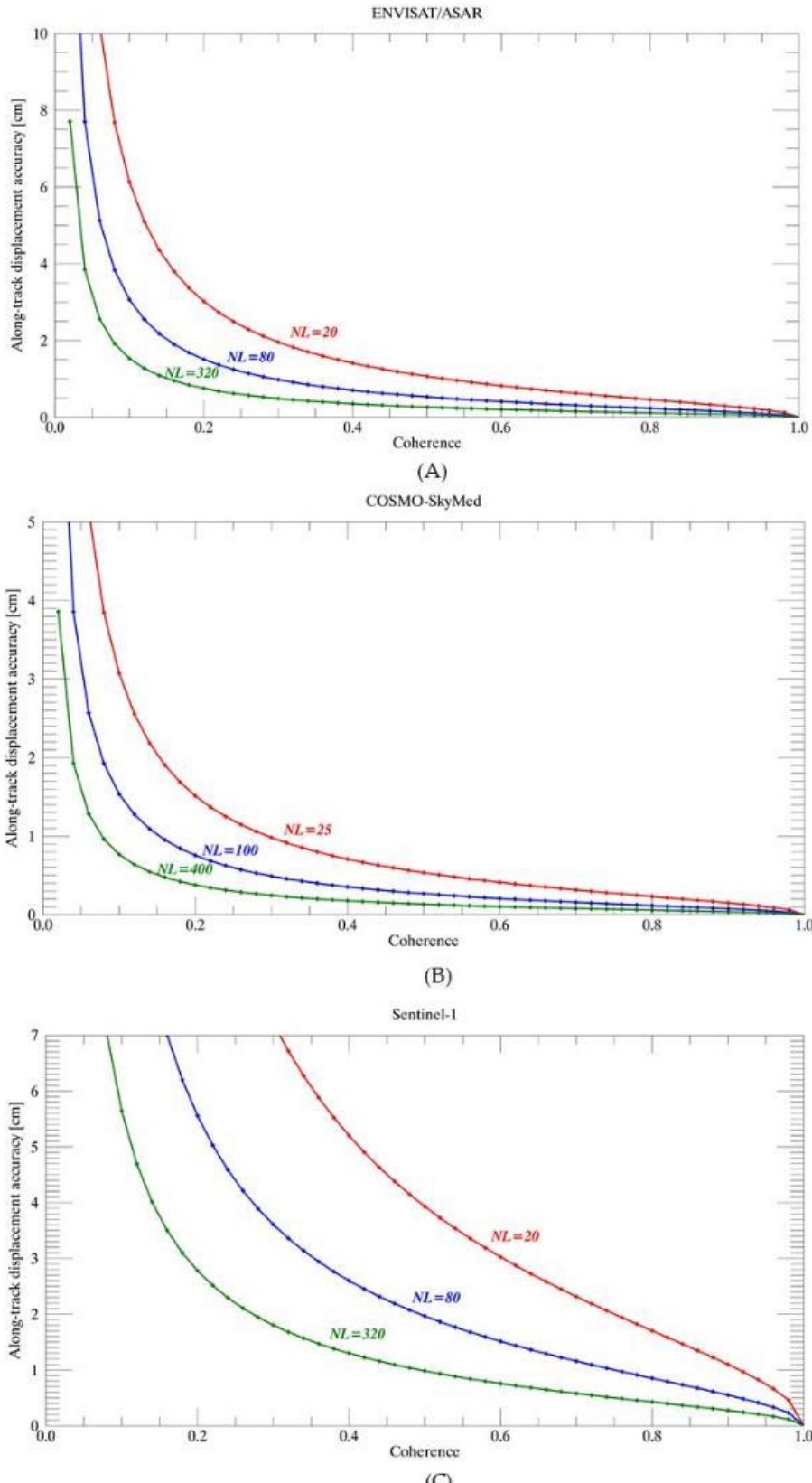

**Figure 14.** Plot of the along-track displacement accuracies vs coherence for the Envisat/ASAR (**A**), COSMO-SkyMed (**B**) and Sentinel-1 (**C**) case using the MAI technique for different NLs (effective look numbers).

**Table 1.** Envisat/ASAR stripmap mode acquisition platform parameters.

| Parameter | Value | Unit |
|---|---|---|
| Wavelength (Centre Frequency) | 0.05624624 | m |
| Band type | C | |
| Pulse repetition frequency | 1.652.4157 | Hz |
| Polarisation options | Single VV, HH or Dual VV+HH, VV+VH, HH+HV | |
| Azimuth antenna size | 11.1 | m |
| Incidence angle range | 15° - 45° | deg. |
| Swath width | 100 | km |
| Resolution | 6(az)x9(rg) | m |
| Azimuth pixel spacing | 4.31 | m |
| Range pixel spacing | 7.8 | m |

**Table 2.** COSMO-SkyMed stripmap (ping-pong) mode acquisition platform parameters.

| Parameter | Value | Unit |
|---|---|---|
| Wavelength (Centre Frequency) | 0.031228381 | m |
| Band type | X | |
| Pulse repetition frequency | 3.554.5024 | Hz |
| Polarisation options | Single VV, HH or Dual VV+HH, VV+VH, HH+HV | |
| Azimuth antenna size | 5.6 | m |
| Incidence angle range | 25° - 50° | deg. |
| Swath width | 30 km | km |
| Resolution | 15(az)x15(rg) | m |
| Azimuth pixel spacing | 2.53 | m |
| Range pixel spacing | 1.56 | m |

**Table 3.** Sentinel-1 operational parameters in interferometric wide (IW) acquisition mode.

| Parameter | Value | Unit |
|---|---|---|
| Wavelength (Centre Frequency) | 0.055465763 | m |
| Band type | C | |
| Pulse repetition frequency | 486.4863 | Hz |
| Polarisation options | Dual HH+VV, VV+HH or Single HH,VV | |
| Azimuth antenna size | 12.3 | m |
| Incidence angle range | 29.1° - 46.0° | deg. |
| Azimuth steering angle | ±0.6° | deg. |
| Swath width | 250 | km |
| Number of sub-swaths | 3 | |
| Resolution | 5(rg)x20(az) | m |
| Azimuth pixel spacing | 14.1 | m |
| Range pixel spacing | 2.3 | m |

Interested readers can find a more comprehensive analysis by Jung et al. in [123].

*3.5. Multiple Aperture Interferometry for the Generation of Along-Track Deformation Time-Series*

SAR interferometric techniques (such as DInSAR and MAI) allow many geophysical events that characterize Earth's surface to be studied in depth through the generation of single deformation maps. However, ground displacement phenomena are processes-characterized by gradual or sudden changes in the Earth's surface elevation over time. In this context, to better characterize the displacement phenomena in an area of interest, an analysis of temporal modifications to ground displacement is

necessary. Over the years, several methodologies have been developed that are useful for extending the use of the differential interferometric SAR techniques to time-monitor displacement phenomena [54–63]. These methods are based on the inversion of properly selected sequences of DInSAR interferograms, allowing the computation of LOS-projected deformation time-series. Two main categories of these techniques are present in literature. The class of the persistent scatterer (PS) methods [54,58] is based on the identification of coherent point-wise targets exhibiting high phase stability over a sequence of DInSAR interferograms that have been generated from a set of SAR data related to the same scene but acquired at different times. Conversely, the class of small baseline (SB) techniques [55,59,63,124] is devoted to the analysis of distributed targets on the ground by processing sequences of small baseline interferograms. More recently, a new advanced method for the characterization of DS targets, called SqueeSAR [61], has also been developed. SqueeSAR uses a statistical approach that exploits phase and amplitude information of a sequence's SAR data to identify very highly coherent DS targets by the use of a maximum-likelihood optimization.

Very recently, temporal analysis methods have been used to extend the MAI technique and retrieve the temporal evolution of the along-track ground displacement. Specifically, a first attempt was made to estimate mean deformation along-track velocity by stacking a sequence $M$ MAI interferograms $\phi = \left[ \phi_{MAI}^0, \phi_{MAI}^1, \ldots, \phi_{MAI}^{M-1} \right]$ generated from a stack of $N$ SAR images. This was done by writing out the following system of equations:

$$
\begin{bmatrix} \Delta t_0 \\ \vdots \\ \Delta t_{N-1} \end{bmatrix} [v] = \begin{bmatrix} \phi_{MAI}^0 \\ \vdots \\ \phi_{MAI}^{N-1} \end{bmatrix}
\tag{27}
$$

where $\phi_{MAI}^i$ and $\Delta t_i$ are the ith MAI interferogram of the sequence and its temporal baseline, respectively, and $v$ is the unknown deformation velocity to estimate. Finally, by resolving the system of Equation (27) in the least-squares (LS) sense, the deformation velocity is retrieved. A novel processing chain to stack a sequence of MAI interferograms was presented in Jo et al. [125] and used to measure the slow-moving azimuthal displacements of the Kilauea volcano on the Big Island, Hawaii. Taking into account that forward- and backward-looking interferograms are affected by noise, they suggested that—instead of estimating the MAI phase from the interferograms and applying the stacking procedure—the multi-temporal forward- and backward-looking residual interferograms could be individually stacked, in order to generate a MAI velocity map as:

$$
v_{MAI} = \frac{l}{4\pi n} \cdot \frac{\left[ \left\{ \sum_{i=1}^N \varphi_{fw,res}^i \right\} - \left\{ \sum_{i=1}^N \varphi_{bw,res}^i \right\} \right]}{\sum_{i=1}^N \Delta t_i}
\tag{28}
$$

where $\varphi_{fw.res}^i$ and $\varphi_{bw.res}^i$ are the interferometric phases of ith residual forward- and backward-looking interferograms, respectively, which are generated by removing a full aperture multi-look differential interferogram from the forward- and backward-looking one; $\Delta t_i$ is the ith time duration; and $N$ is the total number of forward- and backward-looking interferograms.

In particular, the theoretical variance of the estimated deformation velocity map measurements for the MAI stacking was evaluated to be:

$$
\sigma_{v,MAI\_stacking} = \frac{l}{4\pi n} \cdot \frac{\sigma_{\phi,MAI}}{\sqrt{M} \cdot \overline{\Delta t}}
\tag{29}
$$

where $\overline{\Delta t}$ is the average temporal baseline of the whole InSAR distribution. The three-dimensional (3-D) displacement velocity map of the Kilauea Volcano, obtained by combining multi-stacked DInSAR and MAI ground displacement rate maps, is shown in Figure 12 of [125].

Another important temporal study of the Earth's surface displacement via the MAI technique was carried out by Gourmelen et al. [126]. In their work, the authors combined displacement velocity maps that were obtained by applying the DInSAR and MAI techniques to map the ice surface velocity of the Langjokull and Hofsjokull ice caps in Iceland in 1994. In particular, their approach improved the accuracy of the ice flow measurement by a factor of two for E-W, U-D measurements, and up to a factor of 10 for N-S measurements, as compared with a velocity solution based on InSAR and pixel offset (PO).

Another class of studies based on the use of MAI interferograms is the multi-temporal extension of the MAI approach, which is used to generate along-track ground deformation time-series [88].

In this instance, a network of $M$ InSAR data pairs is identified from a set of $N$ SAR images. If $t_M$ and $t_S$ represent the acquisition times of the master and slave images of a given interferogram, respectively, then the MAI phase is expressed via Equation (20) as:

$$\phi_{MAI} = \frac{4\pi}{l} n [\Delta x(t_S) - \Delta x(t_M)] \tag{30}$$

where $\Delta x(t_S)$ and $\Delta x(t_M)$ are the unknown along-track deformations at the master and slave image acquisition times, respectively. This leads to the solution of a system of linear equations that can be expressed, using matrix formalism, as:

$$A \cdot x = \boldsymbol{\phi}_{MAI} \tag{31}$$

where $x$ represents the vector of the unknown time-series deformation, $\boldsymbol{\phi}_{MAI}$ is the vector of the phase components relative to the MAI interferogram and $A$ is the incidence-like matrix of the InSAR network graph, whose jth row is defined as follows:

$$\begin{cases} A\left(j, I_{M,j}\right) = -\frac{4\pi}{l} n \\ A\left(j, I_{S,j}\right) = \frac{4\pi}{l} n \\ otherwise \quad 0 \end{cases} \tag{32}$$

where in $I_M$ and $I_S$ are the $M$-length vectors of the indices of the master and the slave time acquisitions, respectively. The solution of the system of Equation (31) can be obtained in the LS sense as:

$$\widetilde{x} = A^+ \cdot \boldsymbol{\phi}_{MAI} = \left( \left( A^T \cdot A \right)^{-1} \cdot A^T \right) \cdot \boldsymbol{\phi}_{MAI} \tag{33}$$

where $A^+$ indicates the left pseudo-inverse of the matrix $A$. Note that $A$ is a full-rank matrix if all the SAR acquisitions form one single connected set of data. In a more general case where several different subsets of data are present, however, this can lead to a rank-deficient matrix $A$. In this situation, the same strategy adopted in the small baseline subset (SBAS) technique can be applied, which consists of a proper manipulation of the system of Equation (31) and the application of the singular value decomposition (SVD) method (see [57] for additional details). We note that, for typical values of the expected along-track displacement, the unwrapping operation is not required in our case. Furthermore, the phase residuals:

$$e = A \cdot \widetilde{x} - \boldsymbol{\phi}_{MAI} \tag{34}$$

can be used to identify sets of well-processed SAR pixels. In particular, in this paper we propose exploiting the following temporal coherence factor to evaluate the quality performance of the inversion strategy:

$$\Upsilon = \frac{1}{N} \left| \sum_{i=0}^{N-1} exp[je_i] \right| \tag{35}$$

where $e_i$ is the ith phase residual value, and $\Upsilon$ has the same mathematical expression of the temporal coherence factor initially proposed in [127]. However, in this case, the phase residuals are not a result of time-inconsistent phase unwrapping mistakes, but rather phase noise of the MAI interferograms.

The deformation time-series have been obtained by processing a sequence of 14 MAI interferograms generated from a sequence of seven Envisat/ASAR acquisitions of the Afar depression spanning the period between 9 December 2005 and 10 August 2009. The used data are listed in Table 4, and the distribution of the 14 MAI interferograms in the time/perpendicular baseline plane (indicating a Delaunay graph network [128]), is depicted in Figure 15.

**Table 4.** Envisat/ASAR acquisition data of the Afar depression zone. Date format is day/month/year.

| Acquisition n. | Date |
| --- | --- |
| 1 | 19/12/2005 |
| 2 | 27/02/2006 |
| 3 | 04/12/2006 |
| 4 | 10/09/2007 |
| 5 | 28/01/2008 |
| 6 | 25/08/2008 |
| 7 | 10/08/2009 |

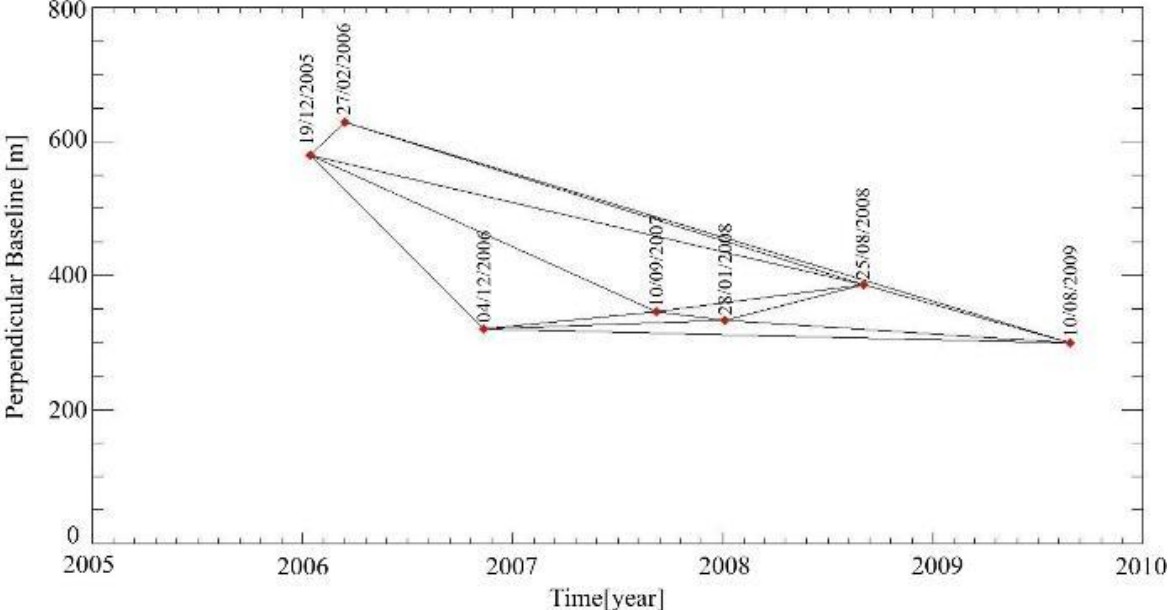

**Figure 15.** Interferometric SAR data pair distribution of the used ASAR/Envisat acquisitions related to the Afar depression area. Red points and black lines indicate the SAR data and the interferograms, respectively.

Figure 16 shows the mean deformation velocity of the Afar area (see Figure 16B) superimposed on a SAR amplitude image of the investigated area, and the relevant time-series of deformation (see Figure 16C–F). Only the deformation values of the coherent, well-processed SAR pixels are shown in the velocity map. They are identified by computing the map of the temporal coherence through Equation (35), as shown in Figure 16A.

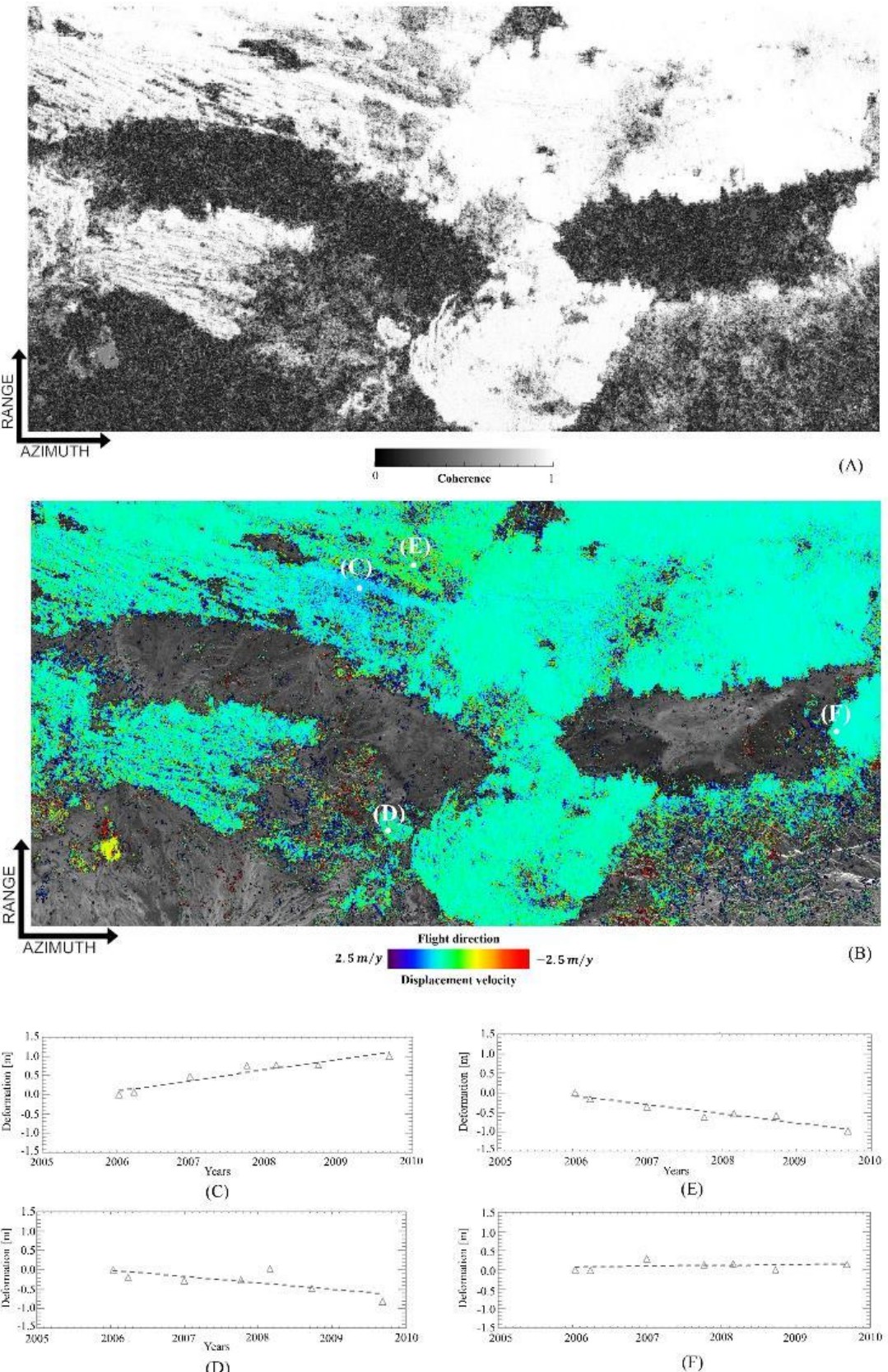

**Figure 16.** (**A**) Coherence map of MAI interferograms. (**B**) Mean deformation velocity map of the Afar area superimposed on a SAR amplitude image of the area. (**C**) Displacement time-series related to the major fault area, (**D**,**E**) related to the medium-magnitude fault area and (**F**) related to the low-magnitude deformation area.

The area in Figure 16B with a sensitive deformation (blue color) is located across the fault, and we highlighted four zones of interest that concerned the time-series analysis (points (B), (C), (D) and (E)). These major fault areas correspond to: (i) a major fault zone whose deformation time-series (Figure 16C) showed an almost constant along-track deformation rate annually, with a leap of about 50 cm in the deformation during the 2005–2006 time-period, when the main seismic event occurred; (ii) a zone of medium-magnitude deformation (Figure 16D–E), where we can perceive opposite behaviors, as if the deformations that characterize the AFAR depression should balance out somehow; and (iii) a zone of low-magnitude deformation (Figure 16F) where the deformation time-series remained constant.

A more quantitative estimate of the accuracy of the achieved along-track deformation time-series can be obtained by applying the basic principles of error noise propagation [129]. In particular, given a full covariance matrix for the vector of MAI phases $C_{MAI} = \sigma_{MAI}^2 I_{M \times M}$, the covariance matrix of the along-track deformation time-series is given by $C_X = \left( A^T \cdot C_{MAI}^{-1} \cdot A \right)^{-1}$, where we have assumed the simplified hypothesis that the MAI phases are independent and uncorrelated, and that the standard deviation of the MAI phase is given by Equation (22). A more extensive treatment is required in the case that matrix $A$ is rank deficient, and a given correlation is assumed between the MAI interferograms. This is a matter for further investigation, which would require a deep understanding of the statistical relationships among the interferograms, as well as a characterization of the studied PS and DS targets on the ground.

More recently, other advanced multi-temporal deformation investigations have been carried out using the MAI technique. Specifically, He et al. [88] performed a temporal analysis of big landslides that had occurred in the Fushun west open-pit mine (FWOPM), China. They mapped the two-dimensional deformation field time-series of landslide phenomena that occurred between 2007 and 2011 by coupling DInSAR-SBAS with MAI-SBAS. They validated the generated time-series (across-track and along-track directions) with GPS displacement measurements over Fushun mine. The time-series obtained by He et al. are presented in Figure 10 of their article [88]. In particular, the results in the two graphics demonstrate that the time-monitoring analysis of the DInSAR and MAI techniques returned information that was highly consistent with the landslide kinematic pattern measured by the GPS sensor in that area.

All of the analyses presented for the generation of along-track ground deformation time-series demonstrate that the key factor is the reduction of decorrelation noise errors. In Liu et al. [130], a very accurate investigation of the effects of these spurious phase components is presented that perturbs the final measurements. Specifically, Liu et al. classified these errors into three categories: (i) random error, (ii) systematic error and (iii) gross error. These categories provide a more comprehensive understanding of the errors in inverting interferometric DInSAR and MAI phases when computing time-series and 3-D maps of deformation of an observed scene.

*3.6. Overview of the Enhanced Spectral Diversity for TOPS Mode SLC Image co-registration*

An enhanced spectral diversity (ESD) approach has also been developed and extensively used for the fine co-registration of pairs of single-look-complex SAR images acquired through the TOPS mode (e.g., IW products). Indeed, the growing demands in having broader swath coverage led to the development of a new advanced imaging mode. In particular, the capability to steer the radar antenna beam along the azimuth direction led to the design of the Terrain Observation by Progressive Scans (TOPS) [131] mode, which is used to achieve wide-swath coverage. As with the conventional ScanSAR mode [132], the TOPS imaging mode achieves wide-swath coverage by switching the antenna beam among range directions from swath to swath (often referred to as sub-swaths) [132].

However, in the TOPS mode (in contrast with the ScanSAR), the antenna beam has an angular velocity $v_{rot}$ that rotates from backward to forward throughout the acquisition with respect to a virtual rotation center (see Figure 17). This allows all targets to be illuminated during the data acquisition duration (burst) within a large portion of the azimuth antenna pattern, thus mitigating the image quality degradation effects such as ambiguity to signal ratio or scalloping effects. However, these benefits are

offset by a worsened azimuthal resolution of SLC-focused images [131], which are now reduced by a factor of $A = \frac{vg}{vs}$, where $vg$ and $vs$ are the ground and the satellite velocities, respectively.

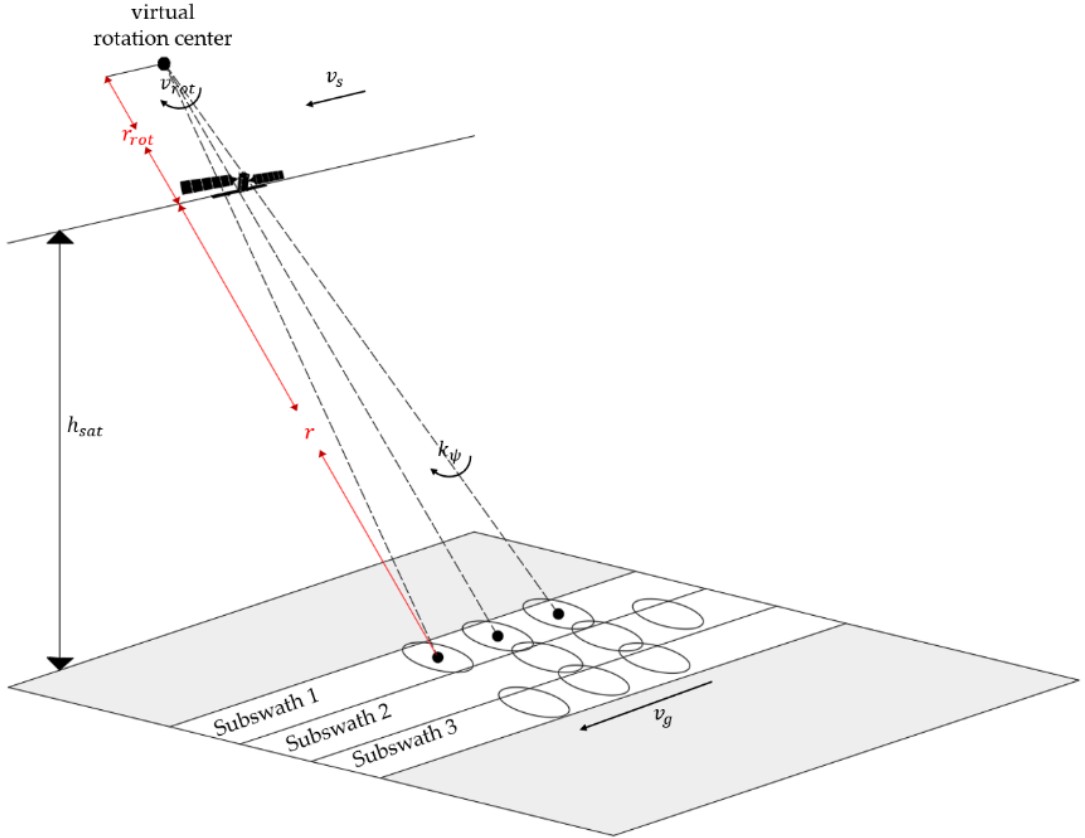

**Figure 17.** Sketch of the Terrain Observation with Progressive Scans (TOPS) acquisition geometry, where $r$ is the sensor-to-target range distance, $r_{rot}$ is the distance between the target on the ground and the virtual rotation center of the acquisition, $v_{rot}$ is the angular velocity of the antenna with respect to a virtual rotation centre,$v_s$ is the satellite velocity, $v_g$ is the ground velocity, $h_{sat}$ is the satellite altitude and $k_\psi$ is the antenna's angular rotation rate.

Let us first introduce the problem by considering two SLC IW images, which are usually acquired over several bursts in three overlapped swaths [79]. The fine co-registration of the two images represents one of the main issues of the Sentinel-1 (S-1) SAR data processing: to achieve this task, it is widely recognized that the azimuth co-registration of the SAR images has to be as accurate as possible to achieve a maximum error not exceeding 0,001 azimuth samples [78,80,81,133,134]. Usually, a conventional geometric co-registration step is performed preceding every SLC burst image.

To introduce the problem, let us consider the two contiguous ith and (i+1)th bursts of the master and slave images—namely $M_i$ and $M_{i+1}$ and $S_i$ and $S_{i+1}$, respectively. In the TOPS case, the Doppler frequency varies linearly for every burst with azimuth time $t_a$—namely, $f_{dop}(t_a) = K_T \cdot t_a$, where $K_T$ is the Doppler frequency rate. Some considerations of TOPS acquisition geometry have led to the conclusion that the Doppler frequency rate within each burst is as follows [81]:

$$K_T = \frac{K_a \cdot K_{rot}}{K_a - K_{rot}} \tag{36}$$

where $K_a = -2v_{eff}^2/\lambda r$ is the Doppler rate of the target, and $K_{rot} = -2v_{eff}^2/\lambda r_{rot} = 2v_s^2 k_\psi/\lambda$. Finally, $k_\psi$ is the angular rotation rate. Note that $v_{eff} = v_g v_s$ is the effective velocity target, where $v_g$ and $v_s$ are the velocities of the ground and the satellite platform, respectively; $r$ and $r_{rot}$ are the sensor-to-target range distance and the distance between the target on the ground and the virtual rotation center of the

acquisition, respectively; and $\lambda$ is the operational wavelength. The (i+1)th and the ith complex burst interferograms—namely, $\Gamma_{i+1}$ and $\Gamma_i$—can be obtained as follows (see Figure 18):

$$\Gamma_{i+1} = M_{i+1} \cdot S_{i+1}^* \tag{37a}$$

$$\Gamma_i = M_i \cdot S_i^* \tag{37b}$$

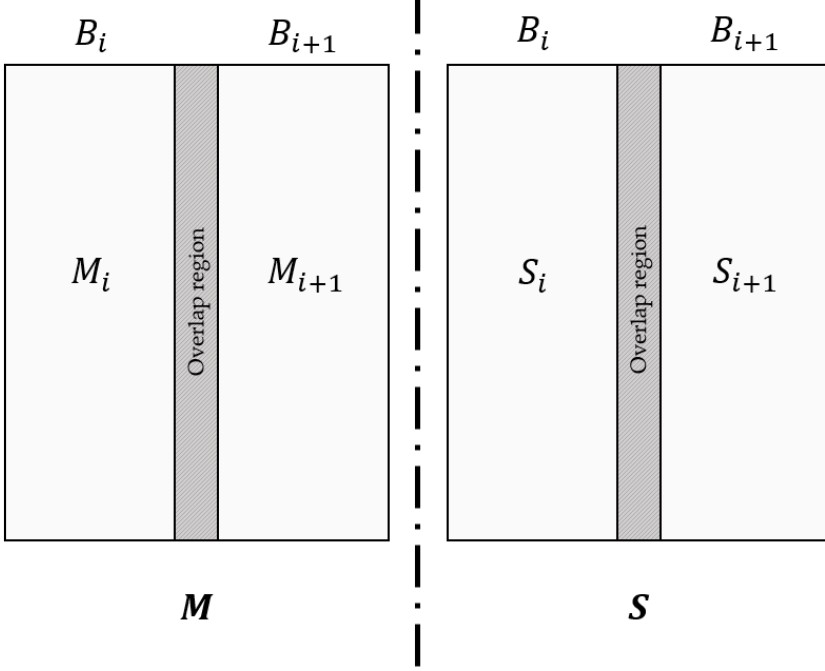

**Figure 18.** Scheme of two TOPSAR master (*M*) and slave (*S*) acquisitions that are related to two contiguous bursts $B_i$ and $B_{i+1}$.

Accordingly, in the overlapped region between the two consecutive ith and (i+1)th bursts, a differential interferogram is formed as

$$\phi_{ovl} = arg\{\Gamma_{i+1} \cdot \Gamma_i^*\} = Wr(\phi_{i+1} - \phi_i) \tag{38}$$

where

$$\phi_{ovl} = \phi_{azDispl} + \phi_{geomErr} + \phi_{Misreg} \tag{39}$$

The first term on the right side of Equation (39), $\phi_{azDispl}$, represents the contribution of the Earth's surface ground displacement along the azimuth direction; the second term, $\phi_{geomErr}$, is the residual geometrical phase due to problems of local co-registration during the geometrical coregistration phase; and the last term, $\phi_{Misreg}$, accounts for the constant (time) azimuth misregistration $\Delta t$ after the co-registration processes. Given a stationary scene and assuming negligible residual geometrical phases, the differential phase over the overlapped region (see Figure 18) is

$$\phi_{ovl} = 2\pi \cdot \Delta f_{ovl} \cdot \Delta t \tag{40}$$

where $\Delta f_{ovl}$ is the Doppler frequency difference between the two bursts at the given SAR pixel in the overlapped region between the two bursts (see Figure 19), which can be computed as follows:

$$\Delta f_{ovl} = K_T \cdot (t_{a,i+1} - t_{a,i}) \tag{41}$$

where $t_{a,i}$ and $t_{a,i+1}$ are the relative times of the ith and i+1th bursts with respect to the starting times of the bursts. The time misregistration is converted in the azimuth misregistration, in terms of the fraction of a pixel, as $\Delta x = \Delta t / \tau$, where $\tau = 1/PRF$ is the time duration of a single azimuth SAR pixel and PRF is the sensor pulse repetition frequency (PRF).

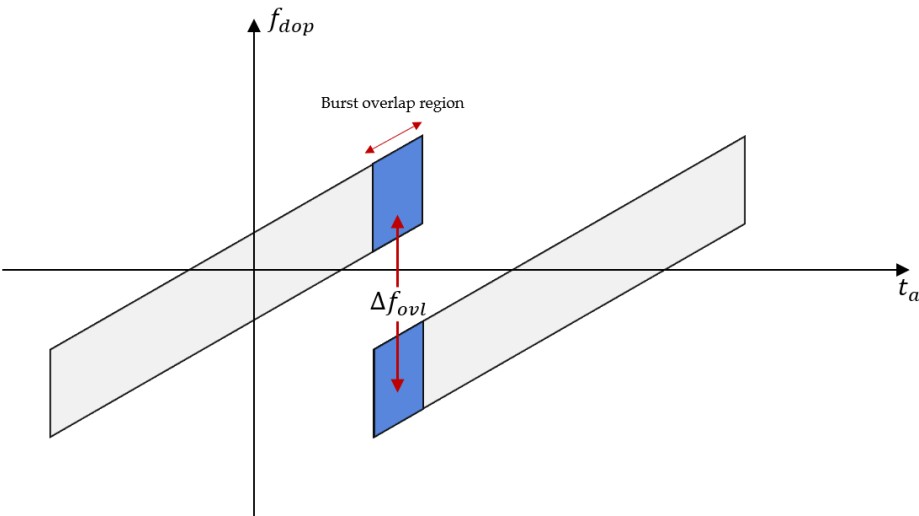

**Figure 19.** Time-frequency diagram of two consecutive focused bursts sketching the enhanced spectral diversity (ESD) approach. The targets located in the overlapped region (shown in blue) have a large spectral separation $\Delta f_{ovl}$.

Therefore, a constant azimuth misregistration leads to a spurious phase ramp along the azimuth direction, which turns out into the presence of phase jumps between consecutive burst interferograms if not adequately compensated for. Note that the phase is measured in the $[-\pi, \pi[$ interval. For typical values of the TOPS mode as implemented in Sentinel-1 system (see the parameters listed in Table 3), the phase in Equation (40) will become ambiguous when the azimuth misalignment (measured in samples) exceeds

$$\Delta x_{max} = \pm \frac{1}{2\tau \cdot \Delta f_{ovl}} = \pm \frac{PRF}{2 \Delta f_{ovl}} \tag{42}$$

which corresponds to $\Delta x_{max} = \pm 0.0119$ when considering the parameters listed in Table 3. The (unknown) azimuth misregistration can usually be estimated by processing some contiguous bursts from the wrapped phase histogram using a simple periodogram [81] as

$$\hat{\Delta} x = \underset{\Delta x}{\mathrm{argmin}} \left\{ Re \left( \sum_p exp \left[ j \left( \phi_{ovl,p} - 2\pi \cdot \frac{\Delta f_{ovl,p}}{PRF} \cdot \Delta x \right) \right] \right) \right\} \tag{43}$$

where $Re(\cdot)$ is the operator that extracts the real part of a complex number. However, if we assume that the preliminary geometrical coregistration is accurate, the measured phase can be considered not ambiguous [134,135], and simply be estimated as

$$\hat{\Delta} x = \frac{PRF \cdot \sum_p \phi_{ovl,p}}{2\pi \cdot \sum_p \Delta f_{ovl,p}} \tag{44}$$

To improve the coherence of the burst overlap interferograms, adaptive azimuth band-filtering is applied to compensate for the different mean Doppler centroids of the bursts and the orbit crossing angles of the two bursts [78].

To show the potential of the ESD approach, let us consider a set of Sentinel-1 SAR data collected over the area of Ridgecrest, California, which was struck by an earthquake in July 2019. The two-burst

interferograms elated to the InSAR data pair between 11 and 23 May 2019 (e.g., in the stationary case) are shown in Figure 20A, whereas Figure 20B shows the phase difference between the two contiguous bursts (before application of the ESD-based co-registration procedure) in the overlapped region (highlighted with a red rectangle). In this case, as is evident, the phase difference only accounts for the residual co-registration errors, as there are no significant along-track deformations present.

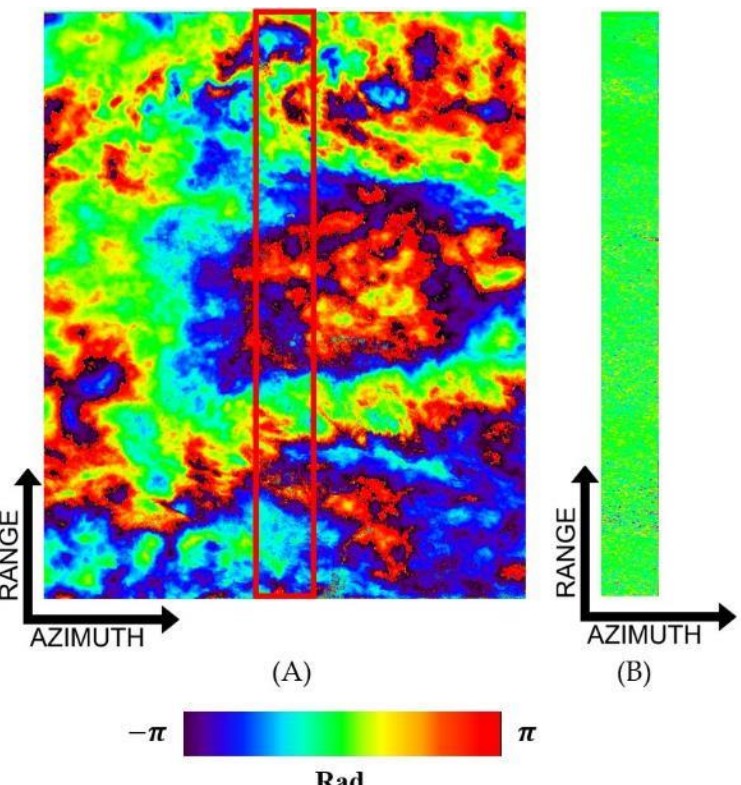

**Figure 20.** (**A**) Two-burst interferogram (stationary case) related to the Sentinel-1 SAR data pairs (11–23 May 2019) of the Ridgecrest area and (**B**) phase difference between the two contiguous bursts in the overlapped region, highlighted with a red rectangle, before ESD application.

The performance of ESD has been evaluated using Terra-SARX [78] and Sentinel-1 data [79,136]. The attainable coregistration accuracy is given by

$$\sigma_{ESD} = \frac{PRF}{2\pi\Delta f_{ovl}} \cdot \sqrt{\frac{3}{NL}} \cdot \frac{\sqrt{1-\gamma^2}}{\gamma} \tag{45}$$

where $\gamma$ is the coherence. Similarly, if the MAI technique is applied to TOPS data, the expected accuracy in the Cramér–Rao limit is [112]

$$\sigma_{MAI} = \frac{PRF}{2\pi\Delta f_{MAI}} \cdot \sqrt{\frac{3}{NL}} \cdot \frac{\sqrt{1-\gamma^2}}{\gamma} \tag{46}$$

The cross-comparison between the expected accuracies in both cases is shown in the plot of Figure 21. The results suggest that only the targets with high coherence have to be selected to estimate the co-registration mistakes. A method to select and exploit the coherent targets has been proposed in [135].

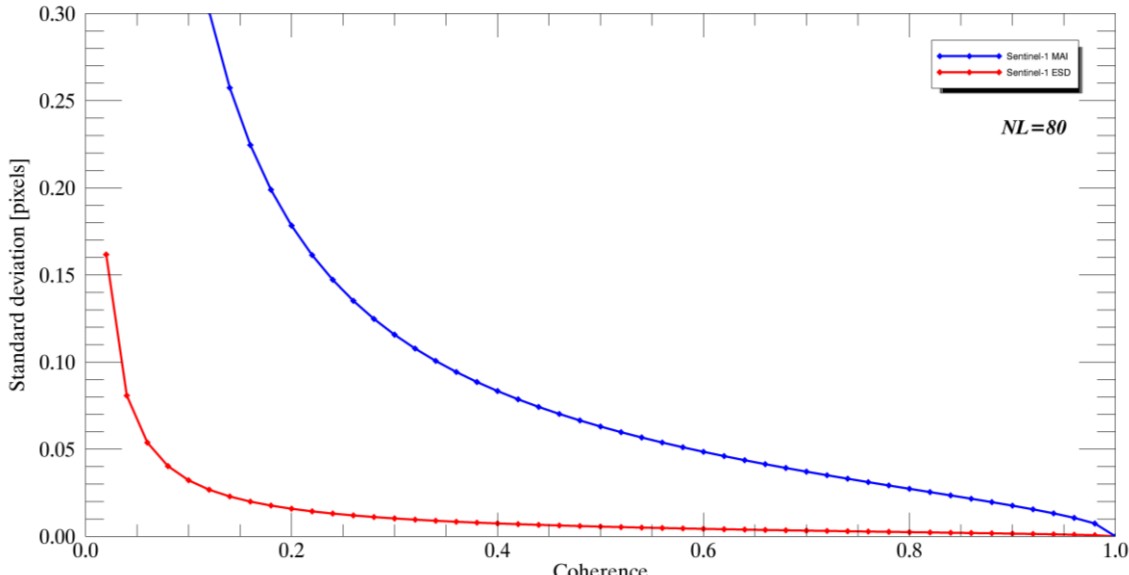

**Figure 21.** Cross-comparison of MAI (blue line) and ESD (red line) root mean squared error (RMSE) vs coherence. NL indicates effective look numbers.

Within the framework of generating surface displacement maps with multi-temporal InSAR methods [55–63], it is crucial to implement proper multi-temporal co-registration procedures that might be efficient for the overall set of InSAR data pairs to be processed. ESD approaches can be successfully applied to estimate the constant misregistration error along the azimuth direction of single SAR data pairs.

If we consider a network composed by *M* multi-temporal InSAR data-pairs, we can compute the *M*-length vector of the azimuthal misregistration—namely, $\Delta X = [\Delta x_1, \Delta x_2, \ldots, \Delta x_M]$—and infer the misregistration errors related to every single SAR image—namely, *X*—by solving the following system of linear equations:

$$\Delta X = A \cdot X \tag{47}$$

in a least-squares or a weighted-least-squares way, where *A* is the incidence-like matrix of the network graph. In [81,134,135,137], different families of InSAR networks are shown. Similarly, as demonstrated for the problem in Section 3.5, the accuracy of the estimated misregistration values for every image is finally obtained as

$$C_X = \left(A^T \cdot C_{\Delta X}^{-1} \cdot A\right)^{-1} \cong \sigma_{\Delta X}^2 \left(A^T \cdot A\right)^{-1} \tag{48}$$

where $\sigma_{\Delta x} \approx 0.001$ [86]. Nevertheless, several challenges are still open that require further effort to perform the coregistration of TOPS SAR data pairs when the scenes are not stationary—that is to say, when a significant azimuth displacement occurs in the analyzed case-study area. In that case, the intra-burst phase difference of the overlapped regions also depends on the azimuthal displacement, thus making estimation of the constant misregistration very challenging and inducing co-registration mistakes in the processed SAR data. To get an idea, Figure 22B shows one bursting interferogram related to the SAR data pairs of 11 May 2019 to 10 July 2019 following the 7.1 Mw earthquake that hit the area.

The burst overlap phase difference estimates the azimuthal displacement [81,123]. In these scenarios, constant misregistration values are usually calculated by excluding the zones with large displacements or by assuming knowledge of a model for the azimuthal ground deformation. Another source of non-stationarity azimuth misregistration is the inhomogeneity in ionospheric propagation delay, as discussed in Section 3.7. Figures 20A and 22A show the pre-seismic and the co-seismic interferograms.

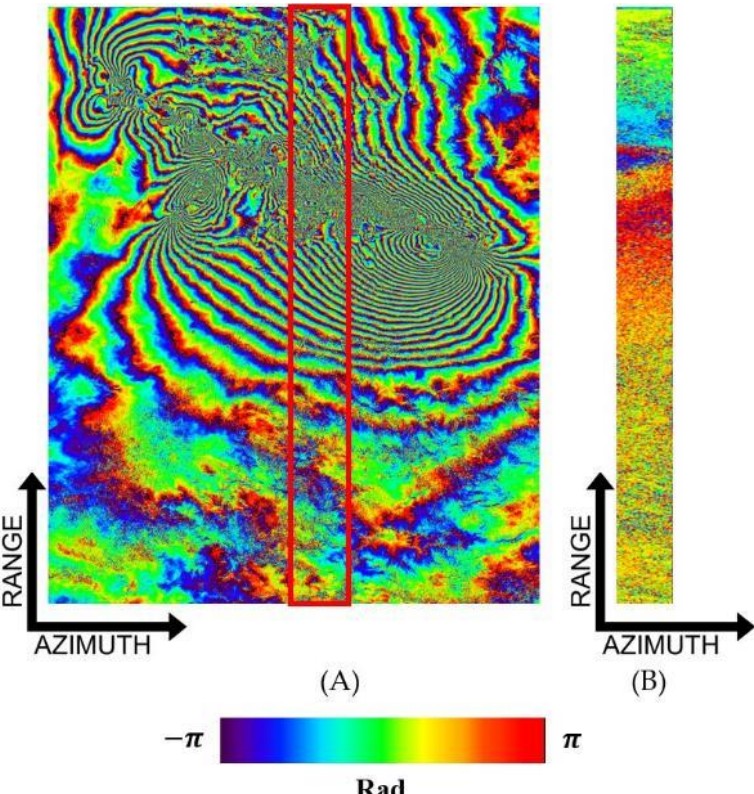

**Figure 22.** (**A**) Two-burst interferogram (non-stationary case) related to the Sentinel-1 SAR data pairs (11 May 2019 to 10 July 2019) of the area of Ridgecrest area (**B**) phase difference between the two contiguous bursts in the overlapped region, highlighted with a red rectangle, before ESD application.

### 3.7. SD Atmospheric Artefacts Retrieval

The inhomogeneity in ionospheric propagation delay causes variable azimuth misregistration that cannot be easily corrected. An important class of phenomena influencing SAR remote sensing measurements concerns the Earth's atmosphere [138], which is the medium through which the electromagnetic radiation propagates. From the InSAR point of view, the uncompensated atmospheric phase screen (APS) represents a factor of great disturbance, because it introduces delays in the measured InSAR phase that result in additional errors in the obtained DInSAR products. Atmospheric disturbances can be referred typically as the result of one of two sources: (i) tropospheric, due to water vapor absorption of electromagnetic radiation, and (ii) ionospheric, due to interactions of electromagnetic radiation with the total electron content (TEC) in the ionosphere. To this aim, several techniques have been developed in recent decades for the mitigation of atmospheric disturbance effects relative to tropospheric and ionospheric effects [54,138,139]. In particular, the absorption effect of water vapor generates an apparent phase signal that masks deformation signals, especially in areas of high relief where topographical phase components are correlated to the atmospheric phase [140–142]. In this context, most of the techniques developed for the correction of the water vapor effects exploit the assumption that atmospheric signals have rapid variations over time (between multiple and temporally different acquisitions of the same scene) and slow variations in space (within a single acquisition) [54,143]. In addition, hybrid methodologies using signal-processing techniques have been developed over the last two decades to attain information about the physical characteristic of water vapor, which are estimated from numerical models of weather forecasting such as those developed by the European Centre for Medium Range Weather Forecast (ECMWF) [140,141,144]. In the case of ionospheric influence, the interactions of electromagnetic waves with electrons contained in the ionosphere can cause path delays on radar signals, particularly with L- and P-band SAR

systems, because the ionospheric phase distortion effects are inversely proportional to the frequency of the used signal. Generally, high TEC and low RADAR frequencies cause significant ionospheric artefacts. In this framework, new MAI-based ionospheric correction techniques have very recently been proposed [83–86] to correct the InSAR phase. These techniques exploit the fact that the azimuth derivative of the ionospheric phase distortion on RADAR interferograms is linearly proportional to the apparent pixel azimuth displacement $\Delta x$:

$$\frac{d\varphi_{ion}}{dx} = \alpha \frac{4\pi}{\lambda} \Delta x \qquad (49)$$

where $\alpha$ is a system- and geometry-dependent factor. Note that $\varphi_{ion}$ can be expressed as

$$\varphi_{ion} = -\frac{4\pi K}{cf} \cdot \frac{1}{cos\theta} \Delta TEC \qquad (50)$$

where K is equal to $40.28 \left(m^3/s^2\right)$, $c$ is the speed of light, $f$ *is* the radar operational frequency and $\Delta TEC$ is the variation of TEC. By using the MAI technique, the estimation of $\Delta x$ can be carried out accurately, as in the study made by Jung et al. [83]. In [83] the authors exploited the potential of the MAI technique to get an accurate estimate and compensation of the ionospheric phase components with a high spatial resolution. Specifically, they applied the method on a pair of ALOS PALSAR (L-band) radar sensor acquisitions affected by strong ionospheric effects. After the generation of the MAI interferograms, several steps were performed to calculate the ionospheric phase components for every azimuth $x$ and range $y$, as follows:

$$\varphi_{ion}(x, y) = \sum_{u=1}^{x} \left[ \left( \alpha \cdot \overline{\varphi}_{MAI}(u, y) + \beta \right) \cdot \Delta_{az} \right] + C(y) \qquad (51)$$

where $\alpha$ and $\beta$ are two parameters determined by a polynomial fitting; $\overline{\varphi}_{MAI} = -\frac{l}{n\lambda}\varphi_{MAI}$ is the scaled MAI phase; $\Delta_{az}$ is the multi-looked azimuth pixel spacing; and C(y) is an integral constant that varies along with the range position. Finally, subtracting the estimated ionospheric phase term from the InSAR algorithm, the corrected InSAR phase $\hat{\varphi}_{InSAR}$ is obtained as

$$\hat{\varphi}_{InSAR}(x, y) = Wr(\varphi_{InSAR}(x, y) - \varphi_{ion}(x, y)) \qquad (52)$$

In Figure 7a of [83], Jung et al. show that the performance obtained by their method in correcting InSAR interferograms was affected by ionospheric effects. Furthermore, in Figure 9a,b of [83], Jung et al. show a 2-D Fourier analysis, relative to the power spectra of the uncorrected and corrected interferograms. Very recently, the same authors of [83] have presented a more-efficient ionospheric correction method that improves the performance of the aforementioned work by using a multivariate regression method [84] to correct both ionospheric and orbital artefacts. Finally, Fatthai et al. [86] very recently proposed an algorithm based on the use of the MAI technique for the time-series estimation of ionospheric phase delay from a stack of SAR acquisitions.

## 4. Generation of Multi-Track 3-D Ground Displacement Time-Series

In this Section, we summarize recent developments in the generation of 3-D ground displacement velocity maps, as well as in the retrieval of 3-D ground deformation time-series. Interested readers can find a more comprehensive overview on this topic in [76]. In our work, the focus is principally on the generation of combined MAI/DInSAR-driven 3-D maps. Moreover, we propose a new algorithm for the generation of 3-D ground displacement time-series, complementing the recently developed minimum acceleration combination (MinA) method [75] (applied for the generation of up–down and east–west deformation components) with the north–south ground displacement time-series estimated by applying the methods shown in Section 3.2.

### 4.1. Overview of the Techniques for the Generation of 3-D Displacement Time-Series

One of the limits of the conventional DInSAR methods is that only the projection of the deformation along the LOS direction can be estimated. Nevertheless, the availability of DInSAR data products (i.e., mean deformation rate and corresponding displacement time-series) computed from SAR data collected from ascending and descending data tracks allows a simple combination of the mean displacement velocity measurements as estimated from (at least) two complementary viewing orbital geometries. By considering the geometry portrayed in Figure 23, it can be demonstrated that the east–west and up–down deformation rate components can be obtained as [68,76]

$$d_{LOS}^{(East)} \approx \frac{d_{LOS}^{(Desc)} - d_{LOS}^{(Asc)}}{2sin(\vartheta)} \tag{53a}$$

$$d_{LOS}^{(Up)} \approx \frac{d_{LOS}^{(Desc)} + d_{LOS}^{(Asc)}}{2cos(\vartheta)} \tag{53b}$$

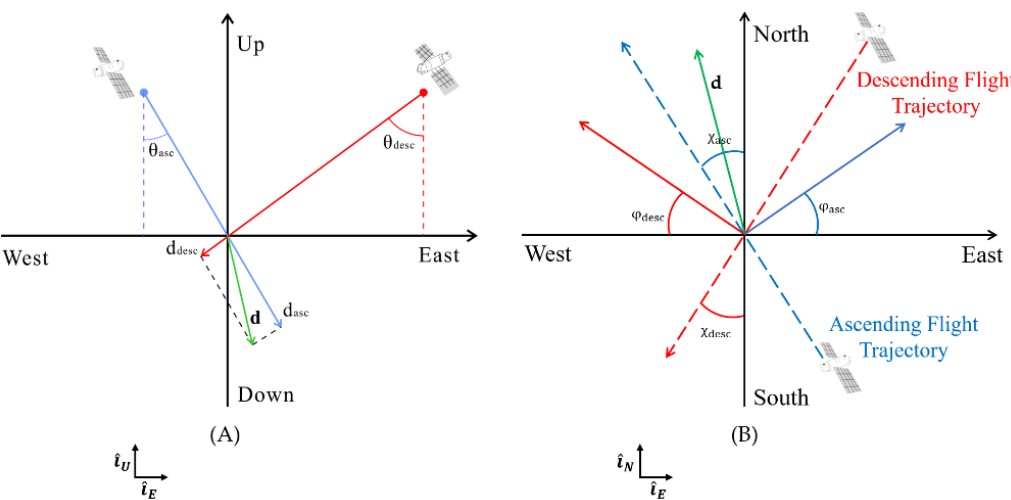

**Figure 23.** Imaging geometries of SAR observations over ascending (blue) and descending (red) orbits in the (**A**) east-vertical and (**B**) east-north planes, respectively.

Equation (53a,b) is obtained by applying simple geometrical relations and assuming that ascending/descending measurements are taken by the same sensor, illuminating the imaged scene with approximately the same (absolute) side-looking angle. Clearly, the sum of the ascending/ descending LOS-projected displacement measurements is related to the vertical components of the ground deformation, whereas the difference of the ascending/descending components gives an estimate of the E-W components of the deformation. A more general relation arises when DInSAR measurements are obtained from complementary viewing angles but from different radar platforms, observing the scene with distinctive side-looking angles, namely, $\theta_{asc}$ and $\theta_{desc}$ (see Figure 16 in [76]):

$$\begin{pmatrix} d_H \\ d_v \end{pmatrix} = \begin{pmatrix} sin(\vartheta_1) & cos(\vartheta_1) \\ -sin(\vartheta_2) & cos(\vartheta_2) \end{pmatrix}^{-1} \cdot \begin{pmatrix} d_{LOS_1} \\ d_{LOS_2} \end{pmatrix} = \frac{\begin{pmatrix} d_{LOS_1}cos(\vartheta_2) - d_{LOS_2}cos(\vartheta_1) \\ d_{LOS_1}sin(\vartheta_2) - d_{LOS_2}sin(\vartheta_1) \end{pmatrix}}{sin(\vartheta_1 + \vartheta_2)} \tag{54}$$

Nonetheless, in the derivation of Equation (54), there is no valuable information related to the Earth's ground deformation along the north–south direction. Indeed, because modern spaceborne radar systems are mounted on satellites that fly nearly polar orbits [41], the north–south (N-S) components of the deformation cannot be reliably measured using conventional DInSAR methods. Several algorithms

have been developed to generate 3-D average displacement maps as well as 3-D displacement time-series by combining multi-satellite/multi-orbit data tracks [65,66,68,72,74]. In our review paper, we mainly concentrate on the MinA technique [75], whose basic rationale is exploited to extend its applicability at the generation of north–south displacement time-series, as presented in Section 4.3. The MinA technique assumes the availability of K independent SAR datasets collected by multiple SAR platforms at the ordered acquisition times $T^{(j)} = \left[ t_0^{(j)}, t_1^{(j)}, \ldots, t_{Qj-1}^{(j)} \right]$, where $Q_j$ is the number of SAR images of the jth SAR dataset. Every single data-track is separately processed, and the LOS-projected displacement time-series are evaluated in correspondence with the location of coherent targets, which are well detected during each data processing step. After geocoding, the DInSAR products are efficiently combined over the common coherent points of the geocoded grid. The combination relies on the application of the mathematical relationships existing between the measured LOS-projection deformations and their corresponding 3-D components (see Figure 23):

$$d_{Los} = \boldsymbol{d} \cdot \hat{\boldsymbol{i}}_{LOS} = sin(\vartheta)cos(\varphi)d_{East-West} - cos(\vartheta)d_{Up-Down} + sin(\vartheta)sin(\varphi)d_{North-South} \tag{55}$$

where $\varphi$ is the radar heading angle (see Figure 23B). In the simplified case, the dependence of the LOS-projected displacement from the north–south components can be neglected. Accordingly, Equation (55) simplifies as follows:

$$d_{Los} = \boldsymbol{d} \cdot \hat{\boldsymbol{i}}_{LOS} \cong sin(\vartheta)d_{East-West} - cos(\vartheta)d_{Up-Down} \tag{56}$$

where we have assumed that $\varphi \cong 0$. The MinA technique relies on the application of Equations (55) and (56) for the generation of 2-D (and 3-D) ground displacement time-series. To this aim, the whole set of chronologically ordered acquisition times—namely, $T = \cup_{j=1}^{K} T^{(j)}$—is considered, and, for every coherent point, the vectors of the velocities among consecutive time acquisitions (see [55]) for the east–west and up–down components—namely, $\boldsymbol{V}_{East-West}$ and $\boldsymbol{V}_{Up-Down}$—are considered as the unknowns of the problem at hand. Accordingly, the following system of linear equations is built (see [75] for the details):

$$\boldsymbol{B} \cdot \begin{bmatrix} \boldsymbol{V}_{East-West} \\ \boldsymbol{V}_{Up-Down} \end{bmatrix} = \begin{bmatrix} d_{LOS}^{(1)} \\ d_{LOS}^{(2)} \\ \vdots \\ d_{LOS}^{(K)} \end{bmatrix} \tag{57}$$

where $\boldsymbol{V}_{East-West}$ and $\boldsymbol{V}_{Up-Down}$ are the vectors of the LOS-projected deformation time-series from the K different, complementary viewing geometries. The matrix $\boldsymbol{B}$, which takes into account the temporal relationships of combined displacement time-series, is expressed as:

$$\boldsymbol{B} = \begin{bmatrix} \boldsymbol{B}^{(1)}sin\vartheta^{(1)}cos\varphi^{(1)} & -\boldsymbol{B}^{(1)}cos\vartheta^{(1)} \\ \vdots & \vdots \\ \boldsymbol{B}^{(K)}sin\vartheta^{(K)}cos\varphi^{(k)} & -\boldsymbol{B}^{(K)}cos\vartheta^{(K)} \end{bmatrix} \tag{58}$$

where $\boldsymbol{B}^{(j)} j = 1, 2, \ldots, K$ is the jth incidence-like matrix (see [55] for further details) of the linear transformation relating the LOS displacement time-series with velocity deformation rates $\boldsymbol{V}_E, \boldsymbol{V}_U$.

To solve the problem, Equation (57) is regularized by imposing the condition that the (unknown) 2-D (E-W, U-D) deformation time-series at a given SAR pixel location has a minimum acceleration [75]. This condition is obtained by adding the following equations to Equation (57):

$$C = \begin{cases} \delta\left(v_{E_{i+1}} - v_{E_i}\right) = 0 \ i = 1, \ldots, Q - 2 \\ \delta\left(v_{U_{i+1}} - v_{U_i}\right) = 0 \ i = 1, \ldots, Q - 2 \end{cases} \tag{59}$$

where δ is a regularization factor. The regularized system of linear equations can be re-written as

$$
\begin{bmatrix} B \\ C \end{bmatrix} \cdot \begin{bmatrix} V_{East-West} \\ V_{Up-Down} \end{bmatrix} = \begin{bmatrix} d_{LOS}^{(1)} \\ d_{LOS}^{(2)} \\ \vdots \\ d_{LOS}^{(K)} \\ 0 \end{bmatrix}
\tag{60}
$$

This system of linear equations is solved in the least-squares (LS) sense as

$$
V = \Upsilon \cdot D
\tag{61}
$$

where $V$ is the (unknown) vector of the LOS-projected E-W and U-P deformation time-series, $\Upsilon$ is the generalized inverse of the matrix $\begin{bmatrix} B \\ C \end{bmatrix}$ and $D$ is the vector of geocoded LOS-displacement time-series relative to the K datasets.

Finally, the east–west and up–down deformation time-series are computed by time integration (pixel by pixel) of the obtained 2-D deformation velocity components. An error budget analysis of the MinA technique has previously been discussed in [145].

A regularized problem that is similar to MinA was proposed within the Multidimensional-SBAS (MSBAS) algorithm [146]. Differently from [75], the combination problem discussed in MSBAS was directly applied to the sequences of unwrapped multiple-track differential SAR interferograms. As opposed to MinA, the MSBAS method relies on searching for a minimum-velocity-norm (MN) solution. Therefore, MSBAS requires the simultaneous inversion of several (a few hundred or more) unwrapped interferograms for the retrieval of 3-D components of deformation.

*4.2. Experimental Results*

Here, we present some results achieved by combining the LOS-projected ground displacement signals collected after applying the multi-temporal SBAS technique to two sets of ascending/descending SAR data. The data were collected by Sentinel-1 radar instruments of the EU Copernicus Sentinels constellation based on the case-study of the area of Ridgecrest in California. The lists of available SAR data are indicated in in Tables 5 and 6 for the ascending and descending data tracks, respectively.

**Table 5.** Sentinel-1 ascending SAR dataset. Date format is day/month/year.

| Acquisition n. | Date |
|:---:|:---:|
| 1 | 11/05/2019 |
| 2 | 23/05/2019 |
| 3 | 16/06/2019 |
| 4 | 28/06/2019 |
| 5 | 10/07/2019 |
| 6 | 22/07/2019 |
| 7 | 03/08/2019 |
| 8 | 15/08/2019 |
| 9 | 27/08/2019 |
| 10 | 08/09/2019 |
| 11 | 02/10/2019 |
| 12 | 14/10/2019 |
| 13 | 26/10/2019 |

Figure 24A,B portrays geocoded LOS-projected mean displacement maps of the area, superimposed on an amplitude image of the investigated area. The geocoding operations have been performed using precise orbital information and a 3-arc-second Shuttle RADAR Topography Mission (SRTM) [147] digital elevation model (DEM).

**Table 6.** Sentinel-1 descending SAR dataset. Date format is day/month/year.

| Acquisition n. | Date |
|:---:|:---:|
| 1 | 05/05/2019 |
| 2 | 17/05/2019 |
| 3 | 29/05/2019 |
| 4 | 10/06/2019 |
| 5 | 22/06/2019 |
| 6 | 04/07/2019 |
| 7 | 16/07/2019 |
| 8 | 28/07/2019 |
| 9 | 09/08/2019 |
| 10 | 21/08/2019 |
| 11 | 02/09/2019 |
| 12 | 14/09/2019 |
| 13 | 08/10/2019 |
| 14 | 20/10/2019 |

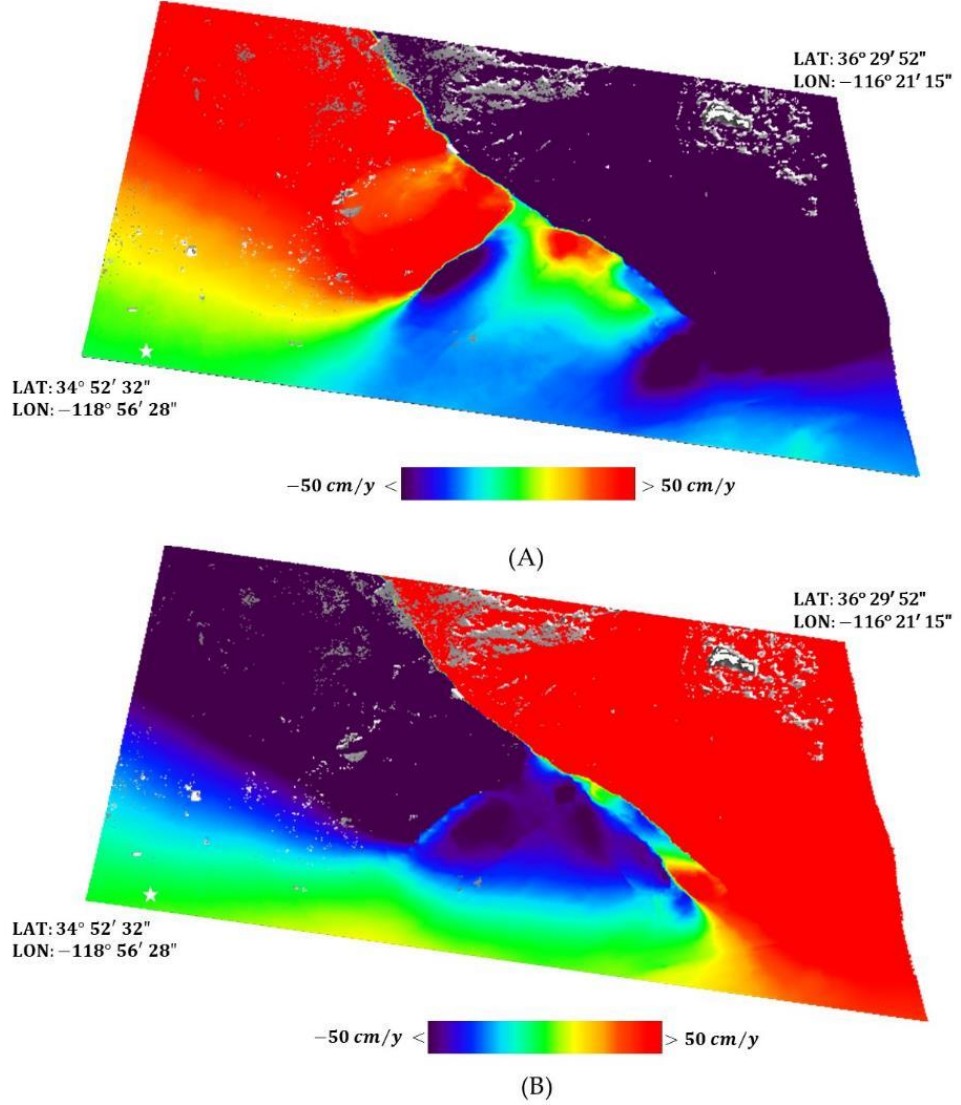

**Figure 24.** Geocoded LOS-projected mean displacement maps of the Ridgecrest area, superimposed on an amplitude image for ascending (**A**) and descending (**B**) orbits, respectively. All the displacement measurements are evaluated with respect to a pixel, identified in (**A,B**) by a white star.

Subsequently, the MinA combination technique was applied to the two sets of geocoded data, to decompose the LOS-projected deformation measurements along the east–west and up–down directions, as shown in Figure 25A,B, respectively

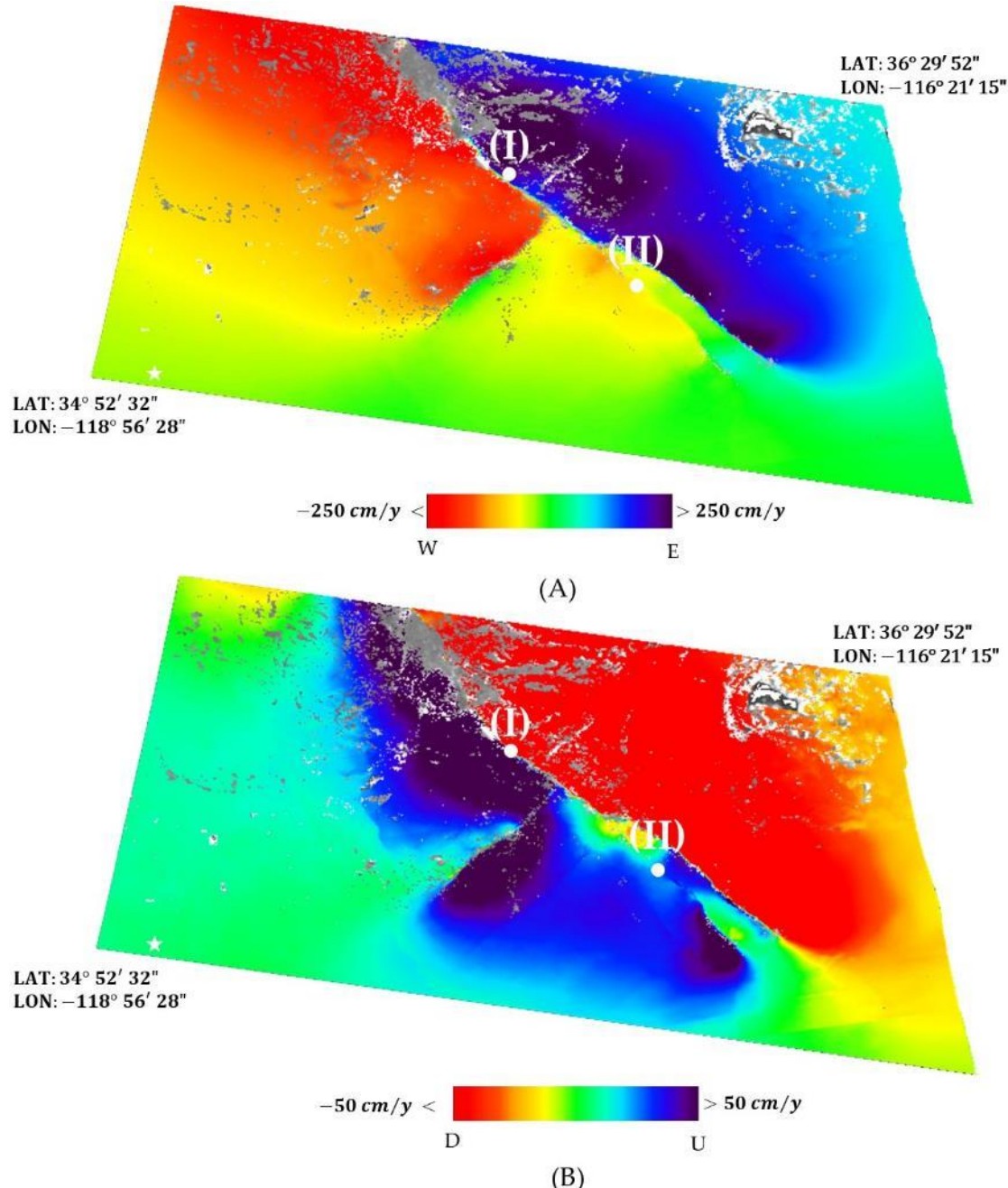

**Figure 25.** Geocoded mean displacement maps along the east–west and the up–down directions of the Ridgecrest area, superimposed on an amplitude image. The labels (I) and (II) in white indicate two points across the fault where the east–west and the up–down displacement time-series have been estimated, respectively. All displacement measurements are evaluated with respect to a pixel, identified in (**A**,**B**) by a white star.

*4.3. Generation of North–South Displacement Maps with MAI Approaches*

In this subsection, we finally propose an adaptation of the MinA multi-temporal technique [75] for the generation of north–south displacement time-series using a combination of K independent sets

of multiple-satellite/multiple-viewing angles. Similar to previous subsections above, we assume here that SAR data were acquired at ordered times $T^{(j)} = \left[ t_0^{(j)}, t_1^{(j)}, \ldots, t_{Qj-1}^{(j)} \right]$, where $Q_j$ is the number of the available SAR scene in the given data track. $T = \cup_{j=1}^{K} T^{(j)}$ is the whole set of ordered acquisition times. Let us also assume that the K sets of data are geocoded over a common grid of points, and $\left\{ d_{az}^{(1)}, d_{az}^{(2)}, \ldots, d_{az}^{(K)} \right\}$ are the K vectors of the azimuthal displacement time-series obtained by applying the MAI approaches described in the previous sections.

By applying simple geometric considerations (see Figure 23B), we can straightforwardly derive that

$$d_{az} = \boldsymbol{d} \cdot \hat{\boldsymbol{i}}_{az} = cos(\chi) \cdot d_{N-s} - sin(\chi) \cdot d_{E-W} \tag{62}$$

where $\chi$ is the inclination angle of the orbit with respect to the N-S direction, which is approximately equal to 6° for the Sentinel-1 case. For each coherent point that is well processed in all of the K independent processing stages, a system of linear equations can be built. In particular, if we consider the velocity deformations as unknowns across contiguous time acquisitions for the east–west and north–south directions, the following system of equations can be written:

$$\boldsymbol{B} \cdot \begin{bmatrix} \boldsymbol{V}_{North-South} \\ \boldsymbol{V}_{East-West} \end{bmatrix} = \begin{bmatrix} \boldsymbol{d}_{az}^{(1)} \\ \vdots \\ \boldsymbol{d}_{az}^{(K)} \end{bmatrix} \tag{63}$$

where $\boldsymbol{V}_{North-South}$ and $\boldsymbol{V}_{Eeast-West}$ are the vectors of the azimuthal projected deformation time-series from the K different, complementary viewing geometries. The matrix **B**, similar to the case accounted for in Section 4.1, takes into account the temporal relationships of combined displacement time-series and, in this case (see the imaging geometry of Figure 23B), is expressed as follows:

$$\boldsymbol{B} = \begin{bmatrix} \boldsymbol{B}^{(1)} cos\chi^{(1)} & -\boldsymbol{B}^{(1)} sin\chi^{(1)} \\ \vdots & \vdots \\ \boldsymbol{B}^{(K)} cos\chi^{(K)} & -\boldsymbol{B}^{(K)} sin\chi^{(K)} \end{bmatrix} \tag{64}$$

Similar to what has already been discussed for the combination of LOS measurements [75], the system of Equation (63) is regularized as

$$\begin{bmatrix} \boldsymbol{B} \\ \boldsymbol{C} \end{bmatrix} \cdot \begin{bmatrix} \boldsymbol{V}_{North-South} \\ \boldsymbol{V}_{East-West} \end{bmatrix} = \begin{bmatrix} \boldsymbol{d}_{az}^{(1)} \\ \boldsymbol{d}_{az}^{(2)} \\ \vdots \\ \boldsymbol{d}_{az}^{(K)} \\ 0 \end{bmatrix} \tag{65}$$

The estimated velocity vector for the north–south components are time-integrated and the N-S ground displacement is computed. Because the projection along the azimuth direction has a very limited accuracy with respect to w.r.t. to the E-W direction, the computed E-W ground displacement time-series (which may be obtained by combining the MAI measurements) definitely has a worse accuracy with respect to w.r.t. obtained using the LOS-projected displacement measurements. The error budget of the N-S displacement time-series can be derived by extending the analysis provided in [145]. The comprehensive theoretical and quantitative analysis of the error budget for the retrieved 3-D ground displacement time-series would require the processing of several independent SAR datasets, as well as a comparison with external ground deformation measurements. This is clearly outside the scope of the current investigation, but remains a matter for future analysis. Figure 26 shows the results of this novel combination method applied to the sets of ascending/descending SAR data relevant to the

area of the Ridgecrest earthquake in California. As a result, we have obtained a map of the north–south mean deformation rate of the area. Finally, we selected two points across the fault, labeled (I) and (II) in Figure 25A,B and Figure 26, and Figure 27 shows the plots of the up–down, east–west and north–south displacement time-series.

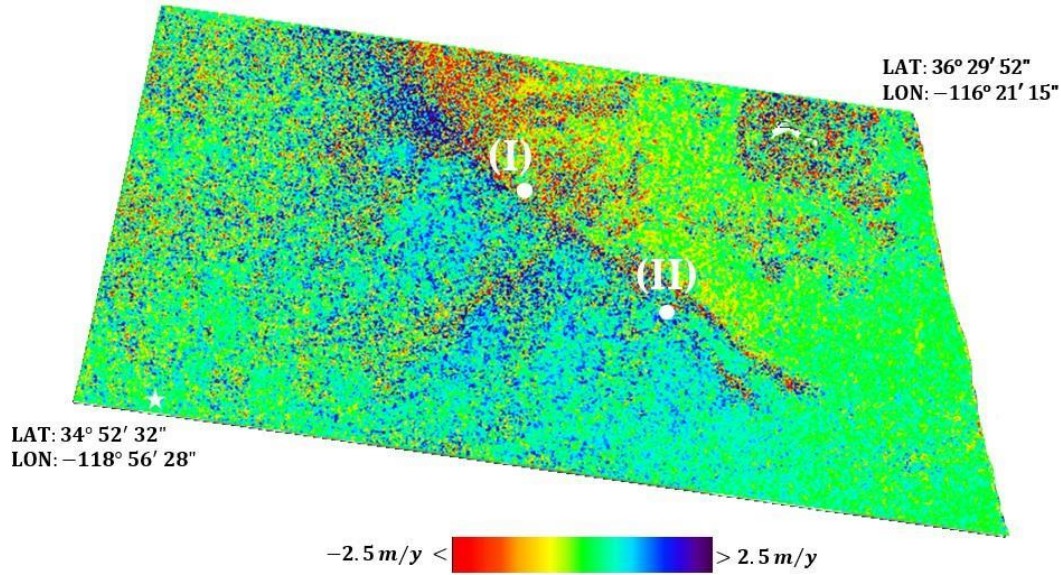

**Figure 26.** Geocoded mean displacement maps along the north–south direction of the Ridgecrest area, superimposed on an amplitude image. The labels (I) and (II) in white indicate two points across the fault where the north–south displacement time-series is estimated. All displacement measurements are evaluated with respect to a pixel, which is identified by a white star.

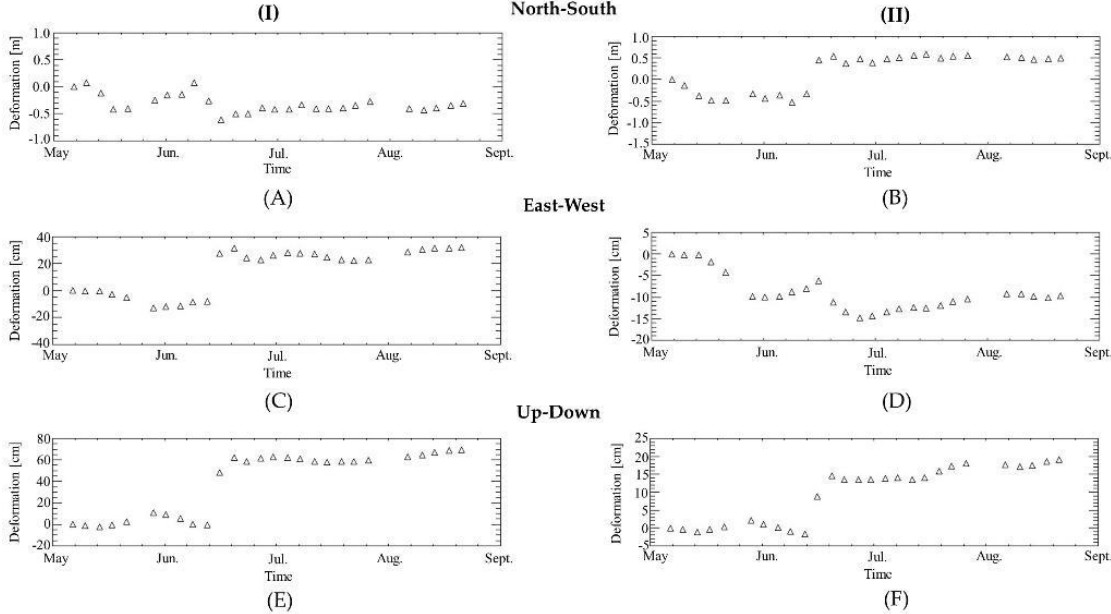

**Figure 27.** North–south, east–west and up–down displacement time-series related to cross-fault points (I) and (II) indicated in Figures 25 and 26.

Additional remarks on the achieved results are now in order. Concerning the Ridgecrest case-study area, the fringe rate in the near proximity of the fault line that was activated by the July 2019 Mw 7.1 earthquake was very high, as is evident from Figure 22A. This led to loss of coherence and unavoidable phase unwrapping errors. The application of conventional InSAR methods, properly

complemented with processing strategies to combine multiple-orbit information, was effective for retrieving the up–down and the east–west ground deformation components. The results shown in Figure 25 prove that most of the deformations were horizontal, with a maximum rate of about 250 cm/year corresponding to an abrupt lateral terrain displacement of about 1 meter. The application of the MAI approach to both ascending and descending Sentinel-1 data tracks also allowed us to obtain an estimate of the north–south displacements according to the generation of relevant deformation time-series (see Figure 26). As shown in Figure 12, we also used one single pair of COSMO-SkyMed SAR images to produce a map of the north–south displacement across the fault line. It should be noted that the magnitude of the retrieved relative N-S deformation across the main seismic event of July 2019 was in general agreement with the deformation time-series of N-S components shown in Figure 26 (with northward and southward deformations of up to 1 meter across the fault). Of course, the accuracy of COSMO-SkyMed and Sentinel-1 MAI-driven measurements were not comparable (see plots shown in Figure 14), as the expected accuracy of MAI measurements with Sentinel-1 data is in the order of 2 cm, with an average coherence value of 0.8. Finally, we would like to remark that most of the experiments were carried out using Sentinel-1 data. The aim was to enhance the strength of spectral diversity approaches for the TOPS mode Sentinel-1 data co-registration, as well as to investigate the potential of the MAI technique applied to TOPS mode SAR data.

## 5. Conclusions

A review of the existing methods used to study large deformation episodes was carried out based on the application of spectral diversity methods for the generation of combined 3-D ground displacement time-series and mean deformation velocity maps. Specifically, we focused on the description of spectral diversity/multiple aperture interferometry methods, and provided a general overview of the theoretical basis of these techniques, addressing their limits and applications, with a particular eye to the use of these technologies for the processing of data coming from new-generation SAR satellites. Several SD techniques were presented for the co-registration of TOPS SAR Sentinel data, the estimation and the removal of the atmospheric phase screen in sequences of SAR data, as well as for the generation of 3-D ground deformation time-series. In the framework of Sentinel-1 SAR data processing, SD and ESD methods would require new adaptations for the analysis of non-stationary scenes where significant along-track displacement signals are present. Furthermore, a novel method combining MAI- and DInSAR-driven data of large deformations was presented, and the relevant results discussed. A quantitative assessment of the presented 3-D methods requires the processing of several datasets. This is a matter for future investigations.

**Author Contributions:** P.M. is the principal investigator. He has reviewed the literature and performed the experiments on real data. A.P. has supervised the work and reviewed the literature related to TOPS Sentinel-1 data with ESD methods. C.S. and G.M. have contributed to the drafting of the manuscript and to the organization of the experimental results. All authors have read and agreed to the published version of the manuscript.

**Funding:** This research received no external funding.

**Acknowledgments:** This work has been supported by the I-AMICA project of structural improvement financing under the National Operational Programme (PON) for "Research and Competitiveness 2007–2013", co-funded by the European Regional Development Fund (ERDF) and National Resources. URBAN-GEO BIG DATA, a Project of National Interest (PRIN) funded by the Italian Ministry of Education, University and Research (MIUR), id. 20159CNLW8, has also partly funded this research study. CSK data of the California area were provided by ASI within the project (ID 699) entitled "Understanding and Predicting Coastal Sea Level Variability Around the United States due to Sea Level Rise and Coastal Subsidence". This research was carried out in the framework of the project PON ARS01_00405 "OT4CLIMA", which was funded by the Italian Ministry of Education, University and Research (MIUR). The authors would like to thank M. Rasulo, F. Parisi and S. Guarino, who have supported this work.

**Conflicts of Interest:** The authors declare no conflict of interest.

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
