# Peer review of "The Multiple Aperture SAR Interferometry (MAI) Technique for the Detection of Large Ground Displacement Dynamics: An Overview"

_remotesensing, doi:10.3390/rs12071189_

Round 1
Reviewer 1 Report
Authors entitle this work an overview of the MAI technique for the detection of large ground displacement dynamics. Indeed this work is huge, near fifty pages in the peer-review format, tons of equations and references and so on. This effort and the overall quality of contents deserves my compliments. Nevertheless, my opinion is that this work requires some refinements. The number of lines is very high for a journal paper; for an overview work this is not a real problem, anyway my opinion is that this work could be compacted a bit without losing clarity. Ad example some concept appear enunciated twice or more. Other advices is to recheck formulas and text errors. In a work so large and rich of not-trivial formulas, errors or undefined variables are very probable. Examples of similar errors are in references where 140 is duplicated in 146 (and so for other references). Finally most figures appear not readable, due to a text too small, especially respect large figures as these; this is a critical point in my opinion. Also captions appear unbalanced, sometimes near excessive, sometimes too synthetic. For example, in figure 14 what is NL?
In my opinion, the most critical points could be solved through a minor revision, and this is my recommendation. Anyway, I have also more struggling comments, indeed not necessary to be accounted for. The structure of this work is correct, because it lead the reader from the basis of SAR interferometry to the proposed specific and advanced technique, but within this structure I suggest collecting all sections describing the used datasets and satellites. In this respect, I found interesting collecting in the same page Tables 1 – 3 for easing the comparison of the three instruments, and having together all the acquisitions of the two study cases. In this respect, the Ridgecrest study case is observed by both Sentinel-1 and CSK; a comparison of the derived products could be better developed. On the other hand, all data of the Afar study case is post-event; this limitation should be properly discussed. Another suggestion is devoted to references. Excluding the duplicated works, the number of references remain very high, so this work could be considered also a sort of review. Nevertheless, I have noticed that some research groups are particularly present, while other important group are missing; moreover, most of the “general” works appear quite old. My suggestion is to complete the literature research, in order to include (or substitute) more recent works and works from other important groups/authors; moreover references should be in some way ordered (now order is not apparent). Finally, I have some doubts on section 3.6: citations do not appear corrected, while the atmospheric artefacts removal approaches do not appear applicate in this work. Moreover, atmospheric effects on SAR signals, including tropospheric ones, is a topic discussed in several work, and in this overview these aspects should be better discussed and cited.
Author Response
See the attached document.

Reviewer 2 Report
The manuscript by Pietro Mastro and colleagues presents an overview of the Multiple Aperture Interferometry technique applied to displacement field of the Earth surface. This methodology, firstly introduced by Bechor and Zebker (2005) in the solid Earth community, is a powerful method to measure ground displacements in the Azimuth direction of the satellite. Therefore, combined with conventional InSAR (i.e. LOS displacement) in ascending and descending tracks, MAI could provide the three dimensional field of the ground displacement. The authors implemented MAI in the Min-A approach (already developed by their group) using not only Sentinel 1, but also other sensors such as ASAR.
I found the topic sound and pertinent for contemporary research; the manuscript is very interesting as an overview. The manuscript is presented in a logical fashion. It is well written –even though I feel that sometimes a native English review is needed. The images are well presented but could be improved (see my minor suggestions below).
The sub aperture idea has been exploited firstly by the Oceanography community, to measure ocean currents (e.g. the Doppler shifts during the focusing, see Chapron et al. J. Geophys. Res., 2005) and by the SAR community to detect ionospheric contributions to the SAR phase (as correctly cited by the authors). The concept of sub-aperture Offset Tracking has been introduced by de Michele et al. (2013, IEEE GRSL) to detect SAR Doppler anomalies during the SAR focusing on a volcanic eruption. Since the manuscript is an overview, I would suggest the authors to open the introduction to these studies above, where the sub-aperture « concept » is applied in general cases.
After these modifications and the ones below, I would suggest the Editor to accept the manuscript for publication.
Minor suggestions for improvement:
Line 69. Pixel Offset : a citation towards the first SAR pixel Offset study in seismotectonics is needed here : Michel, R., Avouac, J.-P., & Taboury, J. 1999, Geophysical Research Letters, 26, 875.
Line 84. ‘The empirical accuracy of PO measurements is limited to about 10 cm’. I see what the authors are trying to say, given the available SAR missions. However, I believe this statement is imprecise here, and unclear for the reader: the accuracy of PO is limited to ~1/10th of the pixel size. Therefore, it depends on the pixel size; it cannot be restrained to « 10 cm ». This is a very important distinction, particularly in cases or airborne SAR or DRONE SAR and Terre-SAR-X spotlight mode.
Line 139. « the crustal Earth ». This term is uncommon; I would suggest « solid Earth » or « the Earth crust ».
237. « meaning a measurement accuracy of millimeters order ». This depends on the SAR band in use. I would suggest to replace with « accuracy of a fraction of the wavelength employed ». Or something similar.
269. The split operation yields a decrease in spatial resolution. Could you write something about this issue here, please?
330 « It has been demonstrated that the measurement accuracy of N‐S displacement that is attainable with PO methods is low. ». This statement is not precise: the accuracy of PO depends on the pixel size (therefore, it is the sensor dependent). The smaller the pixel size, the higher the accuracy.
801. « Azimuth displacement ». This is not clear to me. Do you mean the apparent pixel displacement induced by the ionospheric phase delay? If yes, please modify the sentence.
Figures :
Figure 8. Please add scales and coordinates to all the images
Figure 10. B and C looks like a zoom of A. Is this the case? If yes, you need to state it.
Figure 24. A) I see some unwrapping errors in this interferogram stack. In addition, since there is an earthquake within the data stack -inducing a large offset on the Earth surface- it is not clear what the « mean velocity » physically means.
Figure 25. The unwrapping errors propagate here too. Since you are using a number of interferograms, in a semi-desertic area, is it possible to avoid this kind of errors.
Figures all: put the scale and coordinates
Author Response
See the attached document.

Reviewer 3 Report
A very interesting and up-to-date synthesis.
Author Response
See the attached document.
